# G-REASONER: FOUNDATION MODELS FOR UNIFIED REASONING OVER GRAPH-STRUCTURED KNOWLEDGE

**Linhao Luo**[1], **Zicheng Zhao**[2], **Junnan Liu**[1], **Zhangchi Qiu**[3], **Junnan Dong**[5], **Serge Panev**[6], **Chen Gong**[4], **Thuy-Trang Vu**[1], **Alan Wee-Chung Liew**[3], **Gholamreza Haffari**[1], **Dinh Phung**[1], **Shirui Pan**[3]*

[1]Monash University, [2]Nanjing University of Science and Technology, [3]Griffith University,
[4]Shanghai Jiao Tong University, [5]Tencent Youtu Lab, [6]NVIDIA
`Linhao.Luo@monash.edu`, `s.pan@griffith.edu.au`

 **Project page:** `https://rmanluo.github.io/gfm-rag/`

## ABSTRACT

Large language models (LLMs) excel at complex reasoning but remain limited by static and incomplete parametric knowledge. Retrieval-augmented generation (RAG) mitigates this by incorporating external knowledge, yet existing RAGs struggle with knowledge-intensive tasks due to fragmented information and weak modeling of knowledge structure. Graphs offer a natural way to model relationships within knowledge, but LLMs are inherently unstructured and cannot effectively reason over graph-structured data. Recent graph-enhanced RAG (GraphRAG) attempts to bridge this gap by constructing tailored graphs and enabling LLMs to reason on them. However, these methods often depend on ad-hoc graph designs, heuristic search, or costly agent pipelines, which hinder scalability and generalization. To address these challenges, we present G-reasoner, a unified framework that integrates graph and language foundation models for scalable reasoning over diverse graph-structured knowledge. Central to our approach is Quad-Graph, a standardized four-layer abstraction that unifies heterogeneous knowledge sources into a common graph representation. Building on this, we introduce a 34M-parameter graph foundation model (GFM) that jointly captures graph topology and textual semantics, and is integrated with LLMs to enhance reasoning in downstream applications. To ensure scalability and efficiency, mixed-precision training and distributed message-passing are implemented to scale GFM with more GPUs. Extensive experiments on six benchmarks show that G-reasoner consistently outperforms state-of-the-art baselines, significantly enhances LLM reasoning, and achieves strong efficiency and cross-graph generalization.

## 1 INTRODUCTION

Large language models (LLMs) have demonstrated remarkable reasoning capabilities and serve as the foundation model to solve complex tasks across diverse domains (Achiam et al., 2023; Yang et al., 2025; Liu et al., 2024). However, their effectiveness is often constrained by limitations in accessing up-to-date and domain-specific knowledge (Mousavi et al., 2024; Song et al., 2025b). Recently, retrieval-augmented generation (RAG) (Gao et al., 2023) addresses this challenge by enabling LLMs to reason over external knowledge sources, thereby enhancing their applicability in real-world applications, such as legal judgments (Kang et al., 2024) and medical diagnoses (Jin et al., 2019). While RAG improves access to external knowledge, current RAG approaches struggle with knowledge-intensive reasoning due to the scattered nature of related information (Li et al., 2025b). This requires not only retrieving relevant information but also effectively capturing the associations and structure among knowledge to facilitate reasoning (Jiang et al., 2025).

Graphs provide a natural and flexible representation for modeling the structure and relationships within knowledge (Hogan et al., 2021; Safavi & Koutra, 2021), making them particularly well-suited

---
*Corresponding author.

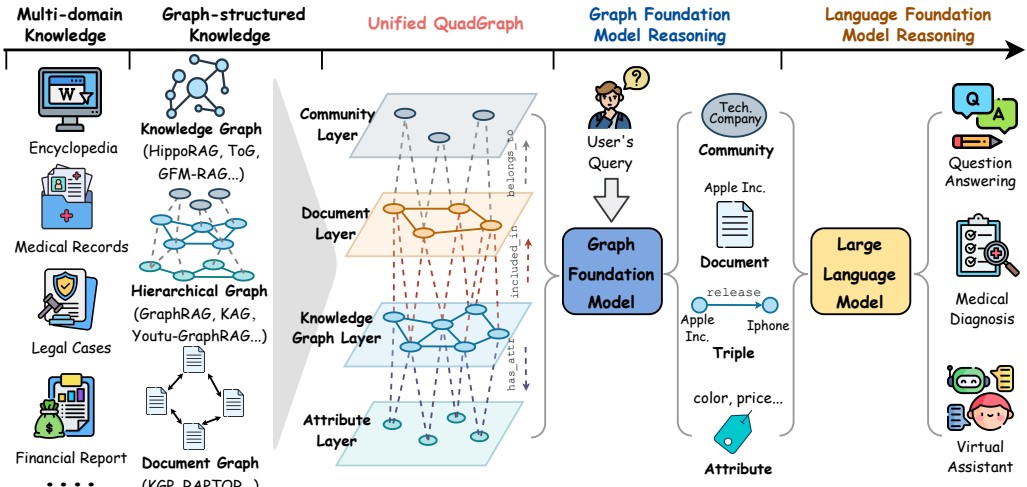

Figure 1: The overall framework of G-reasoner. First, G-reasoner provides a unified graph interface, QuadGraph, that integrates diverse graph-structured knowledge from different domains into a standard format. Then, it adopts a GNN-powered foundation model to jointly reason over the graph-structured knowledge and make versatile predictions. Last, we enhance the LLMs with the graph reasoning results to improve the performance on downstream applications.

for capturing complex knowledge associations to enhance reasoning. However, due to the unstructured nature of LLMs, they struggle to handle graph data (Guo et al., 2023; Jin et al., 2024). This motivates the need for approaches that enhance LLMs to effectively reason over graph-structured knowledge with graph-enhanced retrieval augmented generation (GraphRAG) (Peng et al., 2024; Han et al., 2024).

Existing works in GraphRAG have primarily focused on two components. (1) *Graph construction* focuses on designing a graph structure to effectively organize and capture relationships within the knowledge, such as document graphs (Wang et al., 2024), knowledge graphs (Jimenez Gutierrez et al., 2024), and hierarchical graphs (Edge et al., 2024; Dong et al., 2025). The well-designed graph structure could enhance the retrieval process by providing more context and relationships among knowledge. (2) *Graph-enhanced reasoning* explores to enhance LLMs' ability to reason over these graph structures. For example, HippoRAG (Jimenez Gutierrez et al., 2024) adopts the PageRank algorithm to search over knowledge graphs, ToG (Sun et al., 2024) employs an agent-based approach with tool calling to interact with the graph for reasoning, GNN-RAG (Mavromatis & Karypis, 2025b) leverages graph neural networks (GNNs) to facilitate complex reasoning over graphs.

Despite the effectiveness, existing methods face several limitations. First, they often rely on specific graph structures, which may not generalize well to diverse domains or tasks (Edge et al., 2024; Jimenez Gutierrez et al., 2024). This limits their adaptability and generalizability in real-world applications. Second, intuitive graph search-based methods (Jimenez Gutierrez et al., 2024) may not fully leverage the power of foundation models for reasoning, while agent-based methods (Sun et al., 2024) can be computationally expensive and suffer from high latency. Although GFM-RAG (Luo et al., 2025) proposes a GNN-powered graph foundation model (GFM) with 8M parameters to efficiently reason over graphs, it is still limited to specific knowledge graphs and cannot generalize to other graph structures. Therefore, it is crucial to develop a unified method that can adapt to various graph structures and effectively reason over graph-structured knowledge.

In this paper, we propose G-reasoner, which integrates graph and language foundation models to enable scalable training and generalized reasoning over diverse graph-structured knowledge, as shown in Figure 1. To reason over diverse graph structures, we first define a novel 4-layer graph structure, *QuadGraph*, which unifies heterogeneous graph-structured knowledge into a standardized format. This allows G-reasoner to flexibly adapt to various graph structures. With the unified QuadGraph, we further unleash the power of *graph foundation models* (GFM) powered by GNNs

to jointly reason over the topology and text semantics of the graph. To support large-scale training and reasoning, we implement a mixed-precision training and propose a *distributed message-passing mechanism*, allowing G-reasoner to scale effectively across multiple GPUs and datasets.

Finally, we derive a 34M-parameter GFM that efficiently captures complex relationships and dependencies within the knowledge to make versatile predictions on graphs. The graph reasoning results can be flexibly integrated with LLMs to enhance their reasoning in downstream applications. Experiments on six benchmark datasets demonstrate that G-reasoner achieves superior performance over state-of-the-art baselines and significantly boosts the performance of LLMs on complex reasoning tasks. Moreover, G-reasoner exhibits strong efficiency and generalization capabilities across various graph structures, making it a versatile solution for real-world applications.

The main contributions of this work are summarized as follows:

- We propose G-reasoner, a novel framework that integrates graph and language foundation models to enable unified reasoning over diverse graph-structured knowledge.

- We develop a 34M-parameter graph foundation model that jointly reasons over the graph topology and text semantics, featuring a distributed message-passing mechanism to support large-scale training and reasoning.

- We conduct extensive experiments on six benchmark datasets, demonstrating that G-reasoner achieves superior performance over state-of-the-art baselines and exhibits strong efficiency and generalization capabilities across various graph structures and domains.

## 2 RELATED WORK

**Graph Construction.** Graph construction is key for graph-based reasoning. Early methods like KGP (Wang et al., 2024) use hyperlinks and KNN similarity, but miss semantic associations. RAPTOR (Sarthi et al., 2024) builds hierarchical trees via recursive summarization. GraphRAG (MS) (Edge et al., 2024) use LLMs to extract entities and relations, forming hierarchical graphs with community detection and summarization. LightRAG (Guo et al., 2024), ArchRAG (Wang et al., 2025) and Youtu-GraphRAG (Dong et al., 2025) further enrich graph structures with attributes and documents. HippoRAG 1 & 2 (Jimenez Gutierrez et al., 2024; Gutiérrez et al., 2025) apply OpenIE to induce knowledge graphs capturing factual relationships. Despite their achievements, these methods are typically tailored for specific graph structures, and thus exhibit limited generalizability across different types of graphs. For example, the hierarchical graphs constructed by GraphRAG (MS) (Edge et al., 2024) and LightRAG (Guo et al., 2024) are primarily designed for summarization tasks, and may not be suitable for multi-hop reasoning tasks compared to the knowledge graphs used in HippoRAG (Jimenez Gutierrez et al., 2024). Youtu-GraphRAG (Dong et al., 2025) introduces a vertically unified framework that exploits the graph schema to guide the graph construction.

**Graph-enhanced Reasoning.** Graph-enhanced reasoning seeks to enable LLMs to reason on the graph-structured knowledge and improve their performance on knowledge-intensive applications. HippoRAG (Jimenez Gutierrez et al., 2024) adopts personalized PageRank to support efficient retrieval on knowledge graphs. LightRAG (Guo et al., 2024) employs a dual-level retrieval strategy with both the embedding-based retrieval and graph-based neighborhood expansion. However, these graph search-based methods still fall short of fully exploiting the power of foundation models for reasoning. Agent-based methods, such as ToG (Sun et al., 2024), KAG (Liang et al., 2025), and Youtu-GraphRAG (Dong et al., 2025) employ LLM agents to iteratively interact with graphs to conduct reasoning. Despite the effectiveness, these methods often incur substantial computational costs and suffer from high latency due to the multiple invocations of LLMs. More recent efforts leverage graph neural network (GNNs) to reason over graphs and enhance LLMs Mavromatis & Karypis (2025b); He et al. (2024); Li et al. (2025a). For example, SubgraphRAG (Li et al., 2025a) employs GNNs to encode the graph structure into the node representations, which are then used to retrieve relevant information for LLMs. More recently, GFM-RAG (Luo et al., 2025) proposes a graph foundation model powered by GNNs designed to enable reasoning over different knowledge graphs. However, these approaches remain tailored for specific graphs and cannot generalize well across diverse types of graph structure. More detailed related work can be found in Section A.

## 3 PRELIMINARY

In this section, we formally define the problem of reasoning over graph-structured knowledge with LLMs, which can be unified into a two-stage framework: (1) *graph structure construction* and (2) *graph-enhanced retrieval and LLM reasoning*. Specifically, given a set of documents $\mathcal{D}$, we first extract the knowledge and construct a structured graph $\mathcal{G} = (\mathcal{V}, \mathcal{E})$, such as knowledge graph (Jimenez Gutierrez et al., 2024) and document graph (Wang et al., 2024). The $\mathcal{V}$ denotes the set of nodes (e.g., entity and document) and $\mathcal{E}$ denotes the edges that model the connection between knowledge, facilitating efficient retrieval and reasoning. Based on the constructed graph $\mathcal{G}$ and a user query $q$, we aim to retrieve the relevant knowledge from $\mathcal{G}$ and reason the final answer $a$ with LLMs. The general pipeline can be formulated as:

$$\mathcal{G} = \texttt{GraphConstructor}(\mathcal{D}), \tag{1}$$

$$a = \texttt{LLM}(\texttt{Retriever}(q, \mathcal{G})). \tag{2}$$

## 4 APPROACH

The proposed G-reasoner aims to design a foundation model that unifies the reasoning on diverse graph structures, enabling more effective and efficient reasoning over graph-structured knowledge with LLMs. The overall framework of G-reasoner is illustrated in Figure 1, which consists of three main components: (1) a unified graph interface, QuadGraph, that standardizes diverse graph-structured knowledge from different domains into a unified format; (2) a GNN-powered foundation model that jointly reasons over the graph-structured knowledge and makes versatile predictions; and (3) an LLM-enhanced reasoning that incorporates the graph reasoning results to improve performance on downstream applications. In the following, we will introduce each component in detail.

### 4.1 UNIFIED GRAPH INTERFACE: QUADGRAPH

The real-world knowledge is often complex and multi-relational, which can be naturally represented as graph structures (Hogan et al., 2021; Safavi & Koutra, 2021). To effectively leverage graph-structured knowledge for reasoning, existing methods typically construct different types of graphs based on the specific characteristics of knowledge and requirements of downstream tasks. For example, knowledge graphs (Jimenez Gutierrez et al., 2024) are often used to represent factual information between entities, while document graphs (Wang et al., 2024) are used to capture the relationships between documents based on their content similarity or citation links. However, these methods usually focus on a specific type of graph structure, which limits their applicability to other types of graph-structured knowledge and hinders the generalization of reasoning models.

To address this limitation, G-reasoner proposes a unified graph interface called *QuadGraph* that standardizes diverse graph-structured knowledge from different domains into a unified format. Specifically, we design a 4-layer graph structure that consists of the following layers: (1) *attribute layer* that captures the common attributes of the nodes; (2) *knowledge graph layer* that represents the entities and their relationships as triples, which stores the structured factual knowledge; (3) *document layer* that contains the unstructured textual information, such as documents and passages; and (4) *community layer* that groups related nodes into communities based on their semantic similarity or structural connectivity to provide global level information. As shown in Figure 2, the QuadGraph can effectively unify various types of graph-structured knowledge, such as knowledge graphs (Jimenez Gutierrez et al.,

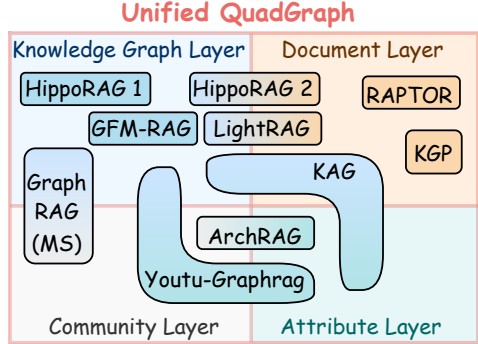

Figure 2: Illustration of QuadGraph for unifying existing graph-structured knowledge.

2024), document graphs (Wang et al., 2024), and hierarchical graphs (Edge et al., 2024; Liang et al., 2025; Dong et al., 2025), into a standard format, facilitating the design of generalizable reasoning models.

**Definition.** The QuadGraph is defined as $\mathcal{G} = (\mathcal{V}, \mathcal{E}, \mathcal{R}, \mathcal{T}, \mathcal{S})$, where $\mathcal{T} = \{\texttt{attribute}, \texttt{entity}, \texttt{document}, \texttt{community}\}$ denotes the set of node types, $\mathcal{R}$ denotes the set of edge types that model the relations between nodes, (e.g., $\texttt{born\_in}, \texttt{city\_of}$) and special relations across layers, (e.g., $\texttt{has\_attribute}, \texttt{included\_in}, \texttt{belongs\_to}$). The edges in the graph are formulated as $\mathcal{E} = \{(v, r, v')|\{t_v, t_{v'}\} \in \mathcal{T}, r \in \mathcal{R}\}$, where $t_v$ denotes the type of node $v$. The $\mathcal{S}$ denotes the set of node semantic features, such as the name of an entity or the text content of a document.

## 4.2 GRAPH FOUNDATION MODEL REASONING

To effectively reason over the unified graph-structured knowledge, G-reasoner proposes a GNN-powered foundation model that jointly reasons over the QuadGraph and makes versatile predictions. Graph neural networks (GNNs) (Mavromatis & Karypis, 2025a; He et al., 2024) have shown great success in reasoning over graph-structured data due to their ability of capturing complex relationships and dependencies between nodes. Recently, GFM-RAG (Luo et al., 2025) proposes a graph foundation model (GFM) for reasoning over knowledge graphs, which demonstrates the effectiveness of GNNs in enhancing LLMs with structured knowledge.

However, GFM-RAG is specifically designed for knowledge graphs and cannot be directly applied to other types of graph-structured knowledge with versatile node types and rich text semantics, such as document graphs or hierarchical graphs. To address this limitation, G-reasoner further unleashes the power of GNNs by designing a more generalizable GFM that (1) synergizes graph topology and text semantics for reasoning and (2) enables versatile predictions on arbitrary node types.

**Synergized Reasoning over Structure and Semantics.** G-reasoner adopts the query-dependent GNN (Galkin et al., 2024; Luo et al., 2025) as the backbone of the GFM, which can capture the complex relationships and dependencies between query and knowledge on the graph. Unlike GFM-RAG (Luo et al., 2025) that only considers the semantics of relations, G-reasoner further incorporates the rich text semantics of nodes $\mathcal{S}$ into the reasoning process.

Given a graph $\mathcal{G}$, we first encode the text features of each node $s_v \in \mathcal{S}$ into node embeddings $\boldsymbol{h}_v \in \mathbb{R}^d$ using a pre-trained text embedding model (e.g., BGE (Chen et al., 2024), Qwen3 Embedding model (Zhang et al., 2025b)). The relation embeddings $\boldsymbol{h}_r \in \mathbb{R}^d$ are also initialized using the same text embedding model to encode the text description of each relation $r \in \mathcal{R}$. With the help of text embeddings, we can effectively capture the semantic information in the graph and unify them into the same embedding space, facilitating the following reasoning.

During the reasoning, the graph $\mathcal{G}$ together with the user's query $q$ are input into the GFM. The model first encodes the query into a query embedding $\boldsymbol{h}_q \in \mathbb{R}^d$ using the same text embedding model to understand the user's intent and align it with the graph knowledge. Then, a $L$-layer query-dependent GNN is applied to jointly reason over the graph topology and text semantics via message-passing and make versatile predictions of each node type, which can be formulated as:

$$\boldsymbol{h}_v^0 = \texttt{Init}(\boldsymbol{h}_v, \mathbf{1}_{v \in \mathcal{V}_q} * \boldsymbol{h}_q), v \in \mathcal{V}, \tag{3}$$

$$\boldsymbol{h}_v^l = \texttt{Update}\left(\boldsymbol{h}_v^{l-1}, \texttt{Agg}\left(\{\texttt{Msg}(\boldsymbol{h}_v^{l-1}, \boldsymbol{h}_r^l, \boldsymbol{h}_{v'}^{l-1})|(v, r, v') \in \mathcal{E}\}\right)\right), l \in [1, L], \tag{4}$$

$$p(v) = \texttt{Predictor}_{t_v}(\boldsymbol{h}_v^L, \boldsymbol{h}_v, \boldsymbol{h}_q), \tag{5}$$

where $\boldsymbol{h}_v^l$ denotes the embedding of node $v$ at the $l$-th GNN layer, the $\texttt{Init}$ function initializes the node embedding by combining the original node embedding $\boldsymbol{h}_v$ and the query embedding $\boldsymbol{h}_q$ if the node $v$ is in the query-related nodes $\mathcal{V}_q$ with a single MLP layer.

At each GNN layer, the $\texttt{Msg}$ function uses DistMult (Yang et al., 2015) to generate the message from the neighbors based on their nodes embeddings $\boldsymbol{h}_v^{l-1}$, $\boldsymbol{h}_{v'}^{l-1}$ and relation embedding $\boldsymbol{h}_r^l$, which are then aggregated by the $\texttt{Agg}$ function (e.g., sum). The $\texttt{Update}$ function updates the target node embedding $\boldsymbol{h}_v^l$ by combining its previous embedding and the aggregated messages using another MLP, and relation embeddings are also updated with a layer-specific MLP, i.e., $\boldsymbol{h}_r^l = g^l(\boldsymbol{h}_r)$.

Finally, a type-specific predictor $\texttt{Predictor}_{t_v}$ is applied to make versatile predictions for each node based on its final embedding $\boldsymbol{h}_v^L$, original text embedding $\boldsymbol{h}_v$, and query embedding $\boldsymbol{h}_q$. The predictor can be designed as a binary classifier for arbitrary node types $t \in \mathcal{T}$, such as entity nodes in the knowledge graph layer or document nodes in the document layer, to predict whether the node is relevant to the query.

**Optimization.** The GFM conducts unified reasoning by integrating the graph topology $(\mathcal{V}, \mathcal{E})$ and text semantics $\mathcal{S}$ in $\mathcal{G}$ to predict the relevance of nodes to the query. The GFM $\theta$ is optimized by maximizing the likelihood of the ground-truth relevant nodes $\mathcal{V}_q^+$, which can be formulated as:

$$\mathcal{O}(\theta) = \sum_{v \in \mathcal{V}_q^+} \log p_\theta(v|q, \mathcal{G}), \tag{6}$$

where the $\mathcal{V}_q^+$ denotes the set of labeled relevant nodes for the query $q$ that can be of arbitrary types $t \in \mathcal{T}$. However, the scarcity of labeled nodes $|\mathcal{V}_q^+| \ll |\mathcal{V}|$ makes it difficult to capture the complex relationships between the query and knowledge on the graph.

To mitigate this challenges, we propose to train the GFM on large-scale datasets with weak supervision by leveraging the abundant unlabeled nodes on the graph. The pre-trained text embedding models (Devlin et al., 2019) have shown strong semantic understanding and can effectively capture the relevance between the query and nodes based on their text features $\mathcal{S}$. Therefore, we propose to leverage the pre-trained text embedding model as a teacher to provide pseudo-labels for all nodes on the graph, which can be formulated as:

$$p_\phi(\mathcal{V}|q, \mathcal{S}) = \texttt{Sigmoid}(\boldsymbol{H}_\mathcal{V}^\top \boldsymbol{h}_q), \tag{7}$$

where $\boldsymbol{h}_q$ denotes the query embedding and $\boldsymbol{h}_v \in \boldsymbol{H}_\mathcal{V}$ denotes the text embeddings of all nodes encoded by the pre-trained text encoder $\phi$, which is frozen during training.

Following the knowledge distillation (Hinton et al., 2015), we train the GFM $\theta$ as a student to minimize the KL divergence between the pseudo-label distribution $p_\phi(\mathcal{V}|q, \mathcal{S})$ and the prediction distribution $p_\theta(\mathcal{V}|q, \mathcal{G})$ over all nodes. As they both follow the Bernoulli distribution, the KL divergence can be efficiently calculated as:

$$D_{\mathrm{KL}}(p_\phi(\mathcal{V}|q, \mathcal{S})||p_\theta(\mathcal{V}|q, \mathcal{G})) = \sum_{v \in \mathcal{V}} p_\phi(v) \log \frac{p_\phi(v)}{p_\theta(v)} + (1 - p_\phi(v)) \frac{1 - p_\phi(v)}{1 - p_\theta(v)}, \tag{8}$$

where $p_\phi(v) = p_\phi(v|q, \boldsymbol{h}_v)$ and $p_\theta(v) = p_\theta(v|q, \mathcal{G})$.

The final unified objective of the GFM training can be formulated as:

$$\mathcal{O}(\theta) = \sum_{v \in \mathcal{V}_q^+} \log p_\theta(v|q, \mathcal{G}) - \lambda D_{\mathrm{KL}}(p_\phi(\mathcal{V}|q, \mathcal{S})||p_\theta(\mathcal{V}|q, \mathcal{G})), \tag{9}$$

where $\lambda$ is a hyper-parameter that balances the two terms. The unified objective not only distill the semantic understanding from the pre-trained text encoder into the GFM but also alleviate the issue of scarce labeled data by leveraging the pseudo-label distribution over the graph. Empirical experiments in Section 5.4 demonstrate the effectiveness of the proposed objectives.

**Large-scale Training and Reasoning.** To enable the generalizable reasoning ability over diverse graph-structured knowledge, G-reasoner is trained on large-scale datasets with weak supervision. Specifically, we collect a large number of query-graph pairs $\{(q_i, \mathcal{V}_{q_i}^+, \mathcal{G}_i)\}_{i=1}^N$ from various domains (Luo et al., 2025), where graphs $\mathcal{G}$ are constructed with diverse graph constructors (e.g., knowledge graphs (Jimenez Gutierrez et al., 2024), document graphs (Gutiérrez et al., 2025), hierarchical graphs (Dong et al., 2025)) and unified into the QuadGraph interface introduced in Section 4.1. The weak supervision $\mathcal{V}_{q_i}^+$ is obtained by labeling the relevant nodes for each query $q_i$, such as answer entities or supporting documents. The GFM is then trained by optimizing the unified objective in eq. (9) over the collected dataset, which can effectively capture the complex relationships between the query and knowledge on the graph and generalize to various types of graph-structured knowledge.

To support large-scale training and reasoning, we first enable *mixed precision training*, yielding an 2.1 times increase in training throughput and a 17.5% reduction in GPU memory. To further scale up the model and graph size, we implement a *distributed message-passing* mechanism that enables distributed training and reasoning across multiple GPUs. Specifically, we partition the full graph into balanced subgraphs using the METIS algorithm (Karypis & Kumar, 1997), with each device storing only a subset of the graph in memory. During the message-passing, each device first aggregates information locally and then exchanges messages with other devices to finalize the node embedding updates. Thus, the memory complexity of G-reasoner per device is $O((|\mathcal{V}|/N) * d)$,

where $N$ denotes the number of devices and $d$ denotes the latent dimension. This design allows G-reasoner to scale effectively to larger graphs and model size by leveraging more GPUs. Detailed implementation and efficiency analysis are provided in Sections C.2 and C.3 and Section 5.5.

### 4.3 LANGUAGE FOUNDATION MODEL REASONING

With the unified QuadGraph and GNN-powered foundation model, G-reasoner can efficiently reason over the graph-structured knowledge and provide versatile predictions for arbitrary node types, such as attributes, entities, documents, and communities. This enables G-reasoner to flexibly select the most relevant information from different layers of the graph at varying granularities, enhancing LLM reasoning and boosting performance in downstream applications.

Specifically, given a user's query $q$, the GFM first reasons over the QuadGraph $\mathcal{G}$ and predicts the relevance score $p(v)$ for each node $v \in \mathcal{V}$. Then, the top-$k$ relevant nodes of each type $\mathcal{V}_q^k = \{\mathcal{V}_{q,t}^k | t \in \mathcal{T}\}$ are selected based on the predicted scores to provide the most relevant information and enhance LLM reasoning, which can be formulated as:

$$\mathcal{V}_{q,t}^k = \texttt{Top-k}\{(p(v)|v \in \mathcal{V}, t_v = t)\}, \tag{10}$$

$$a = \texttt{LLM}(\texttt{Prompt}(q, \mathcal{V}_q^k)), \mathcal{V}_q^k = \{\mathcal{V}_{q,t}^k | t \in \mathcal{T}\}. \tag{11}$$

where $\texttt{Prompt}(\cdot)$ denotes the prompt template that formats the query and information from the selected nodes $\mathcal{V}_q^k$ into a prompt, which is then input into the LLM (e.g., GPT-4 (Achiam et al., 2023), DeepSeek (Liu et al., 2024)) to generate the final answer $a$. Detailed prompt templates are provided in Figure 7.

## 5 EXPERIMENT

In experiments, we aim to answer the following research questions: **RQ1**: Can G-reasoner achieve state-of-the-art performance on reasoning over graph-structured knowledge? **RQ2**: Can G-reasoner effectively generalize across different graph structures? **RQ3**: How do the key components of G-reasoner contribute to its overall performance? **RQ4**: How efficient is G-reasoner in terms of training and inference?

### 5.1 EXPERIMENTAL SETUP

**Datasets.** We first evaluate the effectiveness of G-reasoner on three widely-used multi-hop QA datasets, including HotpotQA (Yang et al., 2018), MuSiQue (Trivedi et al., 2022), and 2WikiMultiHopQA (2Wiki) (Ho et al., 2020), following the settings used in Jimenez Gutierrez et al. (2024); Gutiérrez et al. (2025); Luo et al. (2025) for a fair comparison. To further assess the generalization ability of G-reasoner across domains, we employ three GraphRAG benchmarks: G-bench (Novel) (Xiang et al.,

Table 1: Statistics of the evaluation datasets.

| Dataset | # Query | # Document |
|---|---|---|
| HotpotQA (Yang et al., 2018) | 1,000 | 9,221 |
| MuSiQue (Trivedi et al., 2022) | 1,000 | 6,119 |
| 2Wiki (Ho et al., 2020) | 1,000 | 11,656 |
| G-bench (Novel) (Xiang et al., 2025) | 2,010 | 461 |
| G-bench (Medical) (Xiang et al., 2025) | 2,062 | 2,406 |
| G-bench (CS) (Xiao et al., 2025) | 1,018 | 24,534 |

2025), G-bench (Medical) (Xiang et al., 2025), and G-bench (CS) (Xiao et al., 2025) to evaluate G-reasoner on complex reasoning across medical, novel, and computer science (CS) knowledge. The statistics of the datasets are summarized in Table 1. More details about datasets can be found in Section B.

**Baselines.** We compare with two groups of baselines: (1) *Non-structure methods*: BM25 (Robertson & Walker, 1994), ColBERTv2 (Santhanam et al., 2022), Qwen3-Emb-8B (Zhang et al., 2025b); (2) *Graph-enhanced methods*: RAPTOR (Sarthi et al., 2024), GraphRAG (MS) (Edge et al., 2024), LightRAG (Guo et al., 2024), KAG (Liang et al., 2025), HippoRAG 1 & 2 (Jimenez Gutierrez et al., 2024; Gutiérrez et al., 2025), SubgraphRAG (Li et al., 2025a), G-retriever (He et al., 2024), and GFM-RAG[1] (Luo et al., 2025).

---

[1]We fixed a bug of GFM-RAG in R@k calculation and re-evaluated it in our experiments.

Table 2: QA reasoning performance comparison. GPT-4o-mini is used as the LLM for reasoning.

| Method | HotpotQA | | MuSiQue | | 2Wiki | | G-bench (Novel) | G-bench (Medical) | G-bench (CS) |
|---|---|---|---|---|---|---|---|---|---|
| | EM | F1 | EM | F1 | EM | F1 | ACC | ACC | ACC |
| *Non-structure Methods* | | | | | | | | | |
| None (GPT-4o-mini) (OpenAI, 2024) | 28.6 | 41.0 | 11.2 | 36.3 | 30.2 | 36.3 | 51.4 | 67.1 | 70.7 |
| BM25 (Robertson & Walker, 1994) | 52.0 | 63.4 | 20.3 | 28.8 | 47.9 | 51.2 | 56.5 | 68.7 | 71.7 |
| ColBERTv2 (Santhanam et al., 2022) | 43.4 | 57.7 | 15.5 | 26.4 | 33.4 | 43.3 | 56.2 | 71.8 | 71.9 |
| Qwen3-Emb (8B) (Zhang et al., 2025b) | 53.4 | 67.6 | 31.9 | 44.1 | 57.2 | 63.2 | 56.2 | 70.4 | 73.5 |
| *Graph-enhanced Methods* | | | | | | | | | |
| RAPTOR (Sarthi et al., 2024) | 50.6 | 64.7 | 27.7 | 39.2 | 39.7 | 48.4 | 43.2 | 57.1 | 73.6 |
| GraphRAG (MS) (Edge et al., 2024) | 51.4 | 67.6 | 27.0 | 42.0 | 34.7 | 61.0 | 50.9 | 45.2 | 72.5 |
| LightRAG (Guo et al., 2024) | 9.9 | 20.2 | 2.0 | 9.3 | 2.5 | 12.1 | 45.1 | 63.9 | 71.2 |
| KAG (Liang et al., 2025) | 59.5 | 72.2 | 33.8 | 46.0 | 67.3 | 75.1 | - | - | - |
| HippoRAG (Jimenez Gutierrez et al., 2024) | 46.3 | 60.0 | 24.0 | 35.9 | 59.4 | 67.3 | 44.8 | 59.1 | 72.6 |
| HippoRAG 2 (Gutiérrez et al., 2025) | 56.3 | 71.1 | 35.0 | 49.3 | 60.5 | 69.7 | 56.5 | 64.9 | - |
| SubgraphRAG (Li et al., 2025a) | 44.5 | 57.0 | 25.1 | 35.7 | 62.7 | 69.0 | - | - | - |
| G-retriever (He et al., 2024) | 41.4 | 53.4 | 23.6 | 34.3 | 33.5 | 39.6 | - | - | 69.8 |
| GFM-RAG (Luo et al., 2025) | 56.2 | 69.5 | 30.2 | 49.2 | 69.8 | 77.7 | 58.6 | 72.2 | 72.1 |
| G-reasoner | **61.4** | **76.0** | **38.5** | **52.5** | **74.9** | **82.1** | **58.9** | **73.3** | **73.9** |

**Metrics.** For QA reasoning performance, we use the exact match (EM) and F1 score on multi-hop QA following previous works (Jimenez Gutierrez et al., 2024; Luo et al., 2025) and accuracy (ACC) on G-benchs following their settings (Xiang et al., 2025; Xiao et al., 2025). For retrieval performance, we use document recall@2 (R@2) and recall@5 (R@5) for multi-hop QA and evidence recall (Recall) for G-benchs (Xiang et al., 2025) as evaluation metrics.

**Implementation Details.** We gather the training data from Luo et al. (2025), which consists of 277,839 query samples and 2,972,931 documents, and we construct diverse graph structures using Jimenez Gutierrez et al. (2024); Gutiérrez et al. (2025); Guo et al. (2024); Dong et al. (2025) to train our GFM. We use GPT-4o-mini as the reasoning LLM. More training and implementation details can be found in Section C.

## 5.2 MAIN RESULTS (RQ1)

**QA Reasoning Results.** Table 2 shows QA results on six datasets requiring complex reasoning. G-reasoner consistently outperforms all baselines across these datasets, proving its effectiveness in reasoning over graph-structured knowledge in various domains. Non-structure methods (e.g., BM25, ColBERTv2, Qwen3-Emb) perform poorly on multi-hop QA due to their inability to capture knowledge structure. Graph-enhanced methods (e.g., HippoRAG) generally outperform non-structure methods by leveraging graph structures. However, some approaches relying on specifically designed graphs and heuristic searches (e.g., GraphRAG, LightRAG) struggle to generalize across different datasets and tasks (e.g., G-bench). While the GNN-based GFM-RAG performs well on multi-hop QA, it also underperforms on G-bench datasets, likely due to limited generalization of GNNs across diverse graph structures. In contrast, G-reasoner achieves the best performance across all datasets, demonstrating superior reasoning and generalization capabilities.

To further demonstrate the effectiveness of G-reasoner, we compare it against advanced multi-step (agentic) RAG methods (e.g., IRCoT (Trivedi et al., 2023), ReSearcher (Song et al., 2025a), and Search-R1 (Jin et al., 2025)). From the results in Table 10, we observe that hat G-reasoner consistently outperforms them across all datasets, highlighting its superior ability to leverage graph-structured knowledge for efficient and accurate multi-hop question answering. Unlike agentic RAG approaches, G-reasoner achieves end-to-end reasoning in a single forward pass, offering both improved performance and computational efficiency. The detailed results can be found in Section D.1.

**Retrieval Results.** Table 3 shows retrieval results on multi-hop QA and G-bench datasets. G-reasoner consistently delivers the best performance across all datasets, demonstrating its effectiveness in retrieving relevant information from graph-structured knowledge. Although advanced embedding-based methods (e.g., Qwen3-Emb) perform well by leveraging large-scale pre-training to capture semantic similarity, they still fall short of graph-enhanced approaches on some datasets. This underscores the importance of utilizing graph topology for effective retrieval in complex rea-

Table 3: Retrieval performance comparison. Recall@$k$ (R@$k$) is used for multi-hop QA datasets, and evidence recall (Recall) is used for G-bench (Xiang et al., 2025).

| Method | HotpotQA | | MuSiQue | | 2Wiki | | G-bench (Novel) | G-bench (Medical) |
|---|---|---|---|---|---|---|---|---|
| | R@2 | R@5 | R@2 | R@5 | R@2 | R@5 | Recall | Recall |
| Non-structure Methods | | | | | | | | |
| BM25 (Robertson & Walker, 1994) | 55.4 | 72.2 | 32.3 | 41.2 | 51.8 | 61.9 | 82.1 | 87.9 |
| ColBERTv2 (Santhanam et al., 2022) | 64.7 | 79.3 | 37.9 | 49.2 | 59.2 | 68.2 | 82.4 | 89.5 |
| Qwen3-Emb (8B) (Zhang et al., 2025b) | 74.1 | 88.8 | 46.8 | 62.1 | 66.2 | 74.1 | 82.6 | 92.7 |
| Graph-enhanced Methods | | | | | | | | |
| RAPTOR (Sarthi et al., 2024) | 58.1 | 71.2 | 35.7 | 45.3 | 46.3 | 53.8 | 66.1 | 84.2 |
| GraphRAG (MS) (Edge et al., 2024) | 58.3 | 76.6 | 35.4 | 49.3 | 61.6 | 77.3 | 67.4 | 56.4 |
| LightRAG (Guo et al., 2024) | 38.8 | 54.7 | 24.8 | 34.7 | 45.1 | 59.1 | 79.6 | 82.6 |
| KAG (Liang et al., 2025) | 59.4 | 86.1 | 42.2 | 62.4 | 61.4 | 88.3 | - | - |
| HippoRAG (Jimenez Gutierrez et al., 2024) | 60.1 | 78.5 | 41.2 | 53.2 | 68.4 | 87.0 | 81.2 | 84.0 |
| HippoRAG 2 (Gutiérrez et al., 2025) | 80.5 | 95.7 | 53.5 | 74.2 | 80.5 | 95.7 | 66.2 | 73.6 |
| SubgraphRAG (Li et al., 2025a) | 58.1 | 71.7 | 40.6 | 48.1 | 70.2 | 85.3 | - | - |
| G-retriever (He et al., 2024) | 51.8 | 63.6 | 35.6 | 43.5 | 60.9 | 66.5 | - | - |
| GFM-RAG (Luo et al., 2025) | 75.6 | 89.6 | 43.5 | 57.6 | 79.1 | 92.4 | 75.9 | 82.2 |
| G-reasoner | **85.9** | **97.7** | **54.8** | **74.9** | **81.2** | **98.2** | **87.7** | **93.8** |

Table 4: Generalization of G-reasoner across different graph structures.

| Retriever | Graph Structure | QuadGraph Layer | | | | HotpotQA | | MuSiQue | | 2Wiki | |
|---|---|---|---|---|---|---|---|---|---|---|---|
| | | KG | Doc. | Attr. | Com. | EM | F1 | EM | F1 | EM | F1 |
| Personalized PageRank | HippoRAG | ✓ | - | - | - | 46.3 | 60.0 | 24.0 | 35.9 | 59.4 | 67.3 |
| Embedding+ Graph Search | LightRAG | ✓ | ✓ | - | - | 9.9 | 20.2 | 2.0 | 9.3 | 2.5 | 12.1 |
| G-reasoner | HippoRAG | ✓ | - | - | - | **54.0** | **68.3** | 28.9 | 41.0 | **72.0** | **80.0** |
| | LightRAG | ✓ | ✓ | - | - | 49.7 | 62.0 | 25.3 | 35.9 | 59.4 | 64.4 |
| | Youtu-GraphRAG | ✓ | ✓ | ✓ | ✓ | 52.3 | 65.9 | **30.3** | **42.5** | 69.7 | 77.7 |

soning tasks beyond text semantics. Notably, G-reasoner significantly outperforms existing methods, highlighting the superior ability of our GFM to integrate graph topology and text semantics for efficient retrieval.

## 5.3 Generalization Across Graph Structures (RQ2)

To evaluate the generalization ability of G-reasoner across different graph structures, we conduct experiments using various graph constructors, including HippoRAG (Jimenez Gutierrez et al., 2024), LightRAG (Guo et al., 2024), and Youtu-GraphRAG (Dong et al., 2025), whose statistics are presented in Table 8. The G-reasoner is directly tested on graphs generated by each constructor without further fine-tuning. As shown in Table 4, G-reasoner shows strong generalization ability across different graph structures, consistently outperforming the retrievers specifically designed for each graph type. This demonstrates the robustness and adaptability of G-reasoner in handling diverse graph-structured knowledge for reasoning tasks.

## 5.4 Ablation Study (RQ3)

In this section, we conduct an ablation study to assess the contributions of key components in G-reasoner. We evaluate the impact of (1) *distillation loss* (Distill), (2) *node text semantics* (Text), and (3) *graph foundation model* (GFM) on the performance of G-reasoner. The results are presented in Table 5. Removing the distillation loss leads the performance drops on all datasets, indicating its importance in enhancing the GFM's ability under weak supervision. Ex-

Table 5: Ablation studies of G-reasoner.

| Variant | HotpotQA | | MuSiQue | | 2Wiki | |
|---|---|---|---|---|---|---|
| | R@2 | R@5 | R@2 | R@5 | R@2 | R@5 |
| G-reasoner | **81.1** | **96.9** | **52.1** | **72.4** | **75.6** | **96.1** |
| *w/o* Distill | 77.4 | 96.1 | 50.7 | 71.9 | 75.9 | 96.0 |
| *w/o* Text | 79.4 | 96.3 | 50.0 | 71.9 | 74.6 | 95.2 |
| *w/o* GFM | 11.6 | 19.7 | 3.8 | 7.1 | 4.9 | 9.0 |

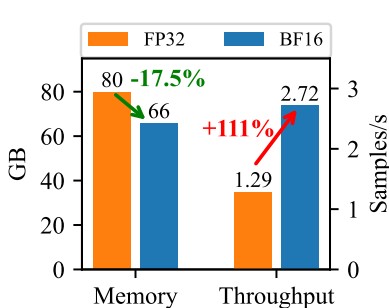

Figure 3: Memory and throughput gain brought by mixed precision training.

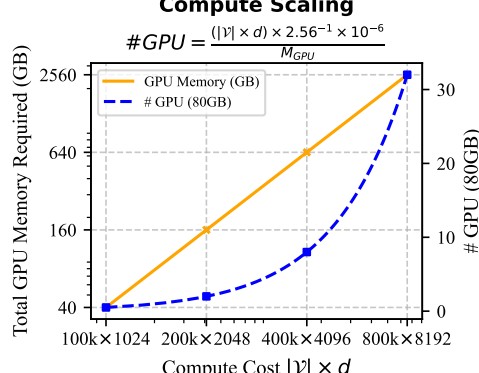

Figure 4: Compute scaling of G-reasoner.

cluding node text semantics also results in performance degradation, highlighting the crucial role of textual information in reasoning tasks. Notably, removing the GFM causes a drastic drop in performance, underscoring its essential role in effectively integrating graph topology and text semantics for reasoning over graph-structured knowledge.

## 5.5 Efficiency Analysis (RQ4)

**Inference Efficiency.** We compare the inference efficiency (time per sample) of G-reasoner on G-bench (CS) (Xiao et al., 2025) with (1) *agent-based*, (2) *graph search*, and (3) *GNN-based methods*. As shown in Table 6, G-reasoner achieves the lowest latency and highest performance among all methods. This demonstrates the efficiency of our method for reasoning over graph-structured knowledge.

**Training Efficiency.** *Mixed precision training* enables G-reasoner to significantly reduce memory usage and improve training throughput. As shown in Figure 3, mixed precision training reduces memory consumption from 80GB to 66GB (-17.5%) and increases throughput from 1.29 to 2.72 samples/s (+111%) on a single A100 GPU. This allows G-reasoner to be trained efficiently on large-scale graph-structured knowledge with limited computational resources.

Table 6: Efficiency and performance comparison on G-bench (CS) (Xiao et al., 2025).

| Method | G-bench (CS) | |
|---|---|---|
| | Time (s) | ACC |
| Agent-based Methods | | |
| KGP (Wang et al., 2024) | 89.4 | 71.9 |
| ToG (Sun et al., 2024) | 70.5 | 71.7 |
| DALK (Li et al., 2024) | 26.8 | 69.3 |
| Graph Search Methods | | |
| GraphRAG (MS) (Edge et al., 2024) | 44.9 | 72.5 |
| LightRAG (Guo et al., 2024) | 14.0 | 71.2 |
| HippoRAG (Jimenez Gutierrez et al., 2024) | 2.4 | 72.6 |
| GNN-based Methods | | |
| G-retriever (He et al., 2024) | 23.8 | 69.8 |
| GFM-RAG (Luo et al., 2025) | 2.0 | 72.1 |
| G-reasoner | **0.2** | **73.9** |

**Compute Scaling.** The compute cost of G-reasoner is defined as $|\mathcal{V}| \times d$ which linearly grows with both the graph node size $|\mathcal{V}|$ and the model's hidden dimension $d$. Thanks to the *distributed message-passing* mechanism, as shown in Figure 4, G-reasoner can efficiently scale to large graphs and larger model sizes with more computational resources. Detailed analysis of compute scaling can be found in Section D.4.

## 6 Conclusion

In this paper, we present G-reasoner, a novel framework that synergizes graph foundation model and language foundation model for reasoning over graph-structured knowledge. With the proposed QuadGraph, G-reasoner unifies diverse graph types into a standardized four-layer graph structure. A GNN-powered graph foundation model is further developed to jointly reason over graph topology and text semantics, enabling versatile prediction on graphs and enhancing LLM reasoning. Extensive experiments on six complex reasoning benchmarks demonstrate that G-reasoner consistently outperforms state-of-the-art baselines, substantially improves LLM reasoning, and exhibits strong efficiency and cross-graph generalization. We believe G-reasoner would pave the road for future research in integrating graph and language foundation models for knowledge-intensive applications.

## ACKNOWLEDGMENTS

This work is partially supported by the DARPA Assured Neuro Symbolic Learning and Reasoning (ANSR) program under award number FA8750-23-2-1016. D Phung is supported by the Australian Research Council (ARC) Discovery Project DP250100262 and DP230101176. S Pan was partly funded by Australian Research Council (ARC) under grants FT210100097 and DP240101547 and the CSIRO – National Science Foundation (US) AI Research Collaboration Program.

## ETHICS STATEMENT

Our research addresses only scientific questions and involves no human subjects, animals, or environmentally sensitive materials. Therefore, we anticipate no ethical risks or conflicts of interest. We are committed to upholding the highest standards of scientific integrity and ethics to ensure the validity and reliability of our findings.

## REPRODUCIBILITY STATEMENT

Our model is clearly formalized in the main text for clarity and thorough understanding. Detailed implementation, including dataset information, baselines, experimental settings, and model configurations, are provided in Sections B, C and 5.1. Experimental settings and baselines have been rigorously verified to ensure fair comparison. Code and pre-trained model weights will be released upon acceptance.

## USAGE OF LLMS

LLMs are used to proofread and polish the writing of this paper. We have carefully reviewed and verified all content generated by LLMs to ensure accuracy and integrity. Any errors or inaccuracies in the final manuscript are solely our responsibility.

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

# Appendix

## Table of Contents

# A  DETAILED RELATED WORK

## A.1  GRAPH CONSTRUCTION

Recently, graph retrieval-augmented generation (GraphRAG) has emerged as a promising approach to leverage structured knowledge to enhance the reasoning capabilities of large language models (LLMs). Nevertheless, suitable graphs are often unavailable for supporting complex multi-hop reasoning task that span across scattered documents. To address this limitation, prior work has explored diverse graph construction strategies tailored to different types of reasoning tasks.

**Document Graph.** KGP (Wang et al., 2024) constructs document graphs using existing hyperlinks and KNN-based similarity, yet the resulting graphs fail to capture the nuanced semantic associations. RAPTOR (Sarthi et al., 2024) builds a hierarchical tree through recursive summarization based on similarities of documents, and SiReRAG (Zhang et al., 2025a) further integrates relatedness with similarity to build tree-like indexing structures for documents.

**Hierarchical Graph.** To better model hierarchical structure, Microsoft GraphRAG (GraphRAG (MS)) (Edge et al., 2024) utilizes LLMs to extract entities and relations from raw texts, and further incorporates community detection with summarization to generate hierarchical graph structure. Building on this line of work, LightRAG (Guo et al., 2024) employs dual-level graph indexing process to facilitate efficient retrieval, whereas Youtu-GraphRAG (Dong et al., 2025) introduces a vertically unified framework that exploits the graph schema to guide the graph construction. Similarly, ArchRAG (Wang et al., 2025) leverages attributed communities (ACs) and introduces an efficient hierarchical retrieval strategy.

**Knowledge Graph.** Beyond document graphs and hierarchical graphs, HippoRAG (Jimenez Gutierrez et al., 2024) and HippoRAG 2 (Gutiérrez et al., 2025) leverage OpenIE techniques to induce knowledge graphs (KGs) that capture the relationships among factual knowledge. To mitigate the noise induced by OpenIE, KAG (Liang et al., 2025) introduces the conceptual semantic reasoning and human-annotated schemas to curate domain expert knowledge.

Despite their achievements, these methods are typically tailored for specific graph structures, and thus exhibit limited generalizability across different types of graphs. For example, the hierarchical graphs constructed by GraphRAG (MS) (Edge et al., 2024) and LightRAG (Guo et al., 2024) are primarily designed for summarization tasks, and may not be suitable for multi-hop reasoning tasks compared to the knowledge graphs used in HippoRAG (Jimenez Gutierrez et al., 2024).

## A.2  GRAPH-ENHANCED REASONING

Graph-enhanced reasoning seeks enable LLMs to reason on the graph-structured knowledge to improve their performance on knowledge-intensive applications.

**Graph Search.** Inspired by hippocampal memory indexing theory, HippoRAG (Jimenez Gutierrez et al., 2024) combines open knowledge graphs with personalized PageRank to support efficient knowledge retrieval on knowledge graphs. Extending on this, HippoRAG2 (Gutiérrez et al., 2025) further incorporates documents into the knowledge graphs, thereby enabling deeper contextual understanding. LightRAG (Guo et al., 2024) employs a dual-level retrieval strategy with both the embedding-based retrieval and graph-based neighborhood expansion to enhance the retrieval performance. However, these graph search-based methods still fall short of fully exploiting the power of foundation models for reasoning.

**Agent-based Reasoning.** Another line of research explores the agent-driven graph reasoning and retrieval. For example, ToG (Sun et al., 2024) employs LLM agents to sequentially interact with graphs and expands relevant reasoning paths for given queries, while ToG2 (Ma et al., 2025) enhances this process by interactively retrieving from both knowledge graphs and documents, thereby achieving context-aware retrieval for reasoning. KAG (Liang et al., 2025) integrates the logical query solver during the agent-based reasoning, which will be called with the query generated by LLMs to perform symbolic reasoning on knowledge graphs. Youtu-GraphRAG (Dong et al., 2025) further proposes an agentic framework that leverages graph schema to guide the LLMs to interact with the graph for reasoning. Despite the effectiveness, these methods often incur substantial computational costs and suffer from high latency due to the multiple invocations of LLMs.

**GNN-based Reasoning.** More recent efforts leverage graph neural network (GNNs) Wu et al. (2020) to reasoning over graph and enhance LLMs. GNN-RAG (Mavromatis & Karypis, 2025b) firstly applies a GNN-based retriever to identify candidate entities for a given question, and then verbalizes entities-induced reasoning paths to support LLMs reasoning. G-retriever (He et al., 2024) combines GNNs with LLMs to enhance the structure understanding of LLMs for reasoning over knowledge graphs. SubgraphRAG (Li et al., 2025a) employs GNNs to encode the graph structure into the node representations, which are then used to retrieve relevant information for LLMs. More recently, GFM-RAG (Luo et al., 2025) proposes a graph foundation model designed to enable reasoning over different knowledge graphs. However, these approaches remain tailored for specific graphs and cannot generalize well across diverse types of graph structure. Although some GFMs have been designed, they primarily focus on graph-related tasks (e.g., node classification (Zhao et al., 2024) and link prediction (Galkin et al., 2024)), making them unsuitable for GraphRAG tasks.

## B    DATASETS DETAILS

We first evaluate the effectiveness of G-reasoner on three widely-used multi-hop QA datasets, including HotpotQA (Yang et al., 2018), MuSiQue (Trivedi et al., 2022), and 2WikiMultiHopQA (2Wiki) (Ho et al., 2020) and three GraphRAG benchmarks: G-bench (Novel) (Xiang et al., 2025), G-bench (Medical) (Xiang et al., 2025), and G-bench (CS) (Xiao et al., 2025). We provide a brief description of each dataset below.

- **HotpotQA** (Yang et al., 2018) is a multi-hop QA dataset that requires reasoning over multiple documents to answer questions. The dataset consists of 97k question-answer pairs, where each question is associated with up to 2 supporting and several distracting documents. The questions are designed to be answerable using multiple pieces of information from the supporting documents.

- **MuSiQue** (Trivedi et al., 2022) is a challenging multi-hop QA dataset with 25k 2-4 hop questions. It requires coherent multi-step reasoning to answer questions that span multiple documents.

- **2WikiMultiHopQA (2Wiki)** (Ho et al., 2020) is a multi-hop QA dataset that requires reasoning over multiple Wikipedia articles to answer questions. The dataset consists of 192k questions, which are designed to be answerable using information from 2 or 4 articles.

- **G-bench (Novel) & G-bench (Medical)** (Xiang et al., 2025) are two domain-specific datasets that are specially designed to evaluate GraphRAG models on both hierarchical knowledge retrieval and deep contextual reasoning. They feature comprehensive datasets with tasks of increasing difficulty, covering fact retrieval, complex reasoning, contextual summarization, and creative generation. G-bench (Medical) collects both domain data from NCCN medical guidelines to provide dense conceptual relationships (e.g., treatment protocols linking symptoms, drugs, and outcomes). G-bench (Novel) collects novels from Gutenberg library to simulate real-world documents with implicit, non-linear narratives.

- **G-bench (CS)** (Xiao et al., 2025) is a dataset that focuses on college-level, domain-specific questions that demand multi-hop reasoning. G-bench (CS) provides comprehensive assessment across the entire GraphRAG pipeline, knowledge retrieval, answer generation, and logical coherence of the reasoning process. It contains 1018 questions in 5 question types spanning 16 topics and a corpus of 7 million words from 20 computer science (CS) textbooks.

In experiments, for multi-hop QA datasets, we adhere existing methods (Jimenez Gutierrez et al., 2024; Luo et al., 2025) to use the same 1,000 samples from each validation set to avoid data leakage. We merge the supporting and distractor passages as the document corpus for graph construction and retrieval. This setup allows us to evaluate the model's ability to retrieve relevant information from a challenging yet controlled environment, reflecting practical scenarios where the model must discern relevant knowledge from a large pool of documents. For G-bench datasets, we follow (Xiang et al., 2025; Xiao et al., 2025) to use the provided test sets and document corpus for evaluation. The statistics of the datasets are summarized in Table 1.

Table 7: Statistics of the training datasets.

| # Query | # Document | # Node | # Relation | # Edge |
|---------|-----------|--------|-----------|--------|
| 277,839 | 2,972,931 | 18,785,120 | 3,920,541 | 77,336,005 |

## C    IMPLEMENTATION DETAILS

### C.1    TRAINING DETAILS

**Training Data.** We gather the training data from Luo et al. (2025), which is based on the training sets of HotpotQA, MuSiQue, and 2Wiki, and construct diverse graph structures to train our GFM. Specifically, the training data consists of 277,839 query samples and 2,972,931 document corpus. Each query is labeled with 2-4 supporting documents. We construct three types of graphs from documents, including knowledge graphs (KG) using HippoRAG (Gutiérrez et al., 2025), knowledge graph + document graph using LightRAG (Guo et al., 2024), and hierarchical graphs using Youtu-GraphRAG (Dong et al., 2025).

The proposed QuadGraph presents a comprehensive schema that integrates four layers: Community, Document, Knowledge Graph, and Attribute, which enables the representation of various graph types within a single framework for training. The construction steps for HippoRAG, LightRAG, and Youtu-GraphRAG are as follows:

- **HippoRAG Graph Construction** (Jimenez Gutierrez et al., 2024; Gutiérrez et al., 2025): HippoRAG contains the knowledge graph layer. We follow the original HippoRAG method to first extract entities, relations, and triples from the document corpus using an LLM-based information extraction approach. Then, we build a knowledge graph layer by connecting entities based on the extracted triples.

- **LightRAG Graph Construction** (Guo et al., 2024): LightRAG employs a dual-level graph indexing process with knowledge graph and document graph. It also first extracts entities and relations from the documents to build a knowledge graph layer. The document layer is constructed by linking documents to the entities they mention.

- **Youtu-GraphRAG Graph Construction** (Dong et al., 2025): Youtu-GraphRAG proposes a hierarchical graph structure with community, document, knowledge graph, and attribute layers, which cover all four layers of QuadGraph. We follow their method to build each layer and connect them accordingly. The knowledge graph is first constructed with schema-bound extraction, and then documents are linked to the entities they mention. Communities are formed by clustering entities with the consideration of both their topographical connectivity and semantic similarity. Attributes are extracted from documents and linked to the corresponding entities.

To ensure efficiency, we split large graphs into smaller subgraphs with around 100k nodes and group the relevant queries for each subgraph during training. The statistics of the training data are summarized in Table 7.

**Model Settings.** The GFM used in G-reasoner is implemented with a 6-layer query-dependent GNN with a hidden dimension of 1024, DistMult message function, and sum aggregation. The relation update function $g^l(\cdot)$ is implemented as a 2-layer MLP. We use the Qwen3-Embedding-0.6B as the sentence embedding model with a dimension of 1024. The total training parameters of the GFM is 34M.

**Training Settings.** The GFM is trained with 16 A100 GPUs (80G) for 10 epochs with a batch size of 2. We use AdamW optimizer with learning rate set to 5e-4. The $\lambda$ for KL divergence is set to 0.01. We also include the ranking loss used in GFM-RAG (Luo et al., 2025) to improve training stability. We apply BFloat16 mixed precision training to reduce memory usage and improve training throughput. The training takes around 7 days to complete. The detailed hyperparameter settings are summarized in Table 9.

**Evaluation Settings.** During the evaluation, for multi-hop QA datasets, we merge the supporting and distractor passages for each query as the document corpus for graph construction and retrieval.

Table 8: Statistics of evaluation graphs constructed by different graph constructor.

| Graph Constructor | | HippoRAG | LightRAG | Youtu-GraphRAG |
|---|---|---|---|---|
| HotpotQA | # Node | 105,256 | 85,130 | 200,533 |
| | # Relation | 24,117 | 54,725 | 7,317 |
| | # Edge | 447,131 | 186,922 | 556,055 |
| MusiQue | # Node | 112,504 | 92,637 | 219,408 |
| | # Relation | 27,973 | 65,404 | 8,471 |
| | # Edge | 464,638 | 210,456 | 636,276 |
| 2Wiki | # Node | 54,898 | 47,361 | 90,258 |
| | # Relation | 10,375 | 101,987 | 2,259 |
| | # Edge | 227,628 | 25,237 | 265,287 |
| G-bench (Novel) | # Node | 29,825 | - | - |
| | # Relation | 11,244 | - | - |
| | # Edge | 108,221 | - | - |
| G-bench (Medical) | # Node | 10,515 | - | - |
| | # Relation | 3,373 | - | - |
| | # Edge | 61,056 | - | - |
| G-bench (CS) | # Node | 217,071 | - | - |
| | # Relation | 36,797 | - | - |
| | # Edge | 1,750,491 | - | - |

Table 9: The detailed implementation and training settings of G-reasoner.

| | | |
|---|---|---|
| GFM | # Layer | 6 |
| | Hidden dim | 1024 |
| | Message | DistMult |
| | Aggregation | Sum |
| | $g^l(\cdot)$ | 2-layer MLP |
| | Sentence embedding model | Qwen3-Embedding-0.6B |
| Training | $\lambda$ | 0.01 |
| | Optimizer | AdamW |
| | Learning rate | 5e-4 |
| | Batch size | 3 |
| | Precision | BFloat16 |
| | Training epochs | 10 |

We use the trained GFM to predict the relevance scores of nodes for each query and select the top-k nodes from each node type to construct the prompt for LLMs. We set $k = 5$ for multi-hop QA datasets, and $k = 10$ for G-bench datasets for fair comparison with existing results. To test the generalizability of G-reasoner across different graph structures, we evaluate G-reasoner on three graph constructors (HippoRAG, LightRAG, Youtu-GraphRAG) for each evaluation dataset. The statistics of the constructed graphs are summarized in Table 8. The results reported in Table 2 and Table 3 are obtained with the graph constructed by HippoRAG.

## C.2 MIXED PRECISION TRAINING

We apply BFloat16 mixed-precision training to reduce memory usage and improve through-put. Mixed precision runs compute-heavy operations (e.g., message-passing) in lower precision while keeping numerically sensitive operations (e.g., reductions) in float32, which typically boosts throughput and reduces memory footprint. This enables training larger models or using larger batch sizes without exhausting GPU memory. However, enabling mixed precision for graph foundation models is non-trivial: we must carefully manage numerical stability during gradient computation in message passing. To address this and fully exploit hardware acceleration, we implemented custom

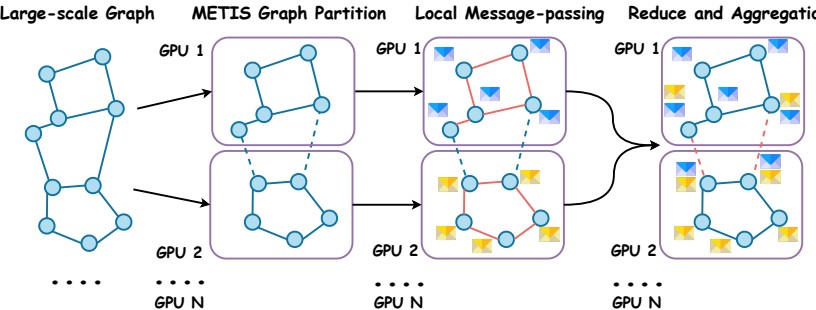

Figure 5: The illustration of distributed message passing in G-reasoner.

Table 10: Performance and efficiency comparison with multi-step RAG methods.

| Method | HotpotQA | | | MuSiQue | | | 2Wiki | | |
|---|---|---|---|---|---|---|---|---|---|
| | EM | F1 | Time / sample (s) | EM | F1 | Time / sample (s) | EM | F1 | Time / sample (s) |
| IRCoT | 45.5 | 58.4 | 1.146 | 19.1 | 30.5 | 1.152 | 35.4 | 45.1 | 2.095 |
| R1-searcher | 61.2 | 73.8 | 0.532 | 34.7 | 48.4 | 0.588 | 58.3 | 71.1 | 0.713 |
| Search-R1 | 60.8 | 74.3 | 0.496 | 37.4 | 53.2 | 0.603 | 54.6 | 68.7 | 0.652 |
| G-reasoner | **61.4** | **76.0** | **0.114** | **38.5** | **52.5** | **0.125** | **74.9** | **82.1** | **0.058** |

CUDA backward kernels for our custom relational message-passing that accumulate gradients in float32, mitigating precision loss while preserving the speed benefits of lower-precision compute.

### C.3 DISTRIBUTED MESSAGE-PASSING

With the customized message-passing CUDA kernels, the memory complexity of GFM is reduced to $O(|\mathcal{V}| * d)$ (Zhu et al., 2021). According to the neural scaling law observed for GFM (Luo et al., 2025) the performance of GFM improves as we increase the model size (i.e., hidden dimension) and the training data size (i.e., number of nodes in graphs). However, the memory consumption of GFM still grows linearly with the number of nodes and hidden dimension, which limits the scalability of GFM on a single GPU. To address this, we implement a distributed message-passing algorithm that partitions the graph across multiple GPUs and performs message-passing in parallel. As shown in Figure 5, we partition the nodes of the graph into $N$ disjoint sets using the METIS algorithm (Karypis & Kumar, 1997) and assign each set to a different GPU. During the message-passing, each GPU computes the messages for its assigned nodes and exchanges the messages with other GPUs as needed. This allows us to scale GFM to larger graphs and model sizes by leveraging more GPU resources. Different from the existing distributed GNN training methods (e.g., PyG (Fey et al., 2025), DGL (Wang et al., 2019)) that use graph sampling, our distributed message-passing algorithm enables full-graph training. This is crucial for preserving the graph structure and ensuring effective reasoning with GFM by passing messages across the entire graph.

## D ADDITIONAL EXPERIMENT

### D.1 COMPARISON WITH MULTI-STEP RAG METHODS

To demonstrate the effectiveness of G-reasoner, we compare the performance with advanced multi-step RAG methods (e.g., IRCoT (Trivedi et al., 2023), ReSearcher (Song et al., 2025a), and Search-R1 (Jin et al., 2025)). From the results in Table 10, we observe that G-reasoner outperforms these advanced RAG systems across all three datasets, demonstrating its effectiveness in multi-hop question answering tasks. While these RAG systems, powered by powerful LLM agents, are designed for iterative retrieval and reasoning, they often lack the ability to effectively capture and leverage the rich relational structure present in graph-structured knowledge. In contrast, G-reasoner 's integration of GFM-based graph reasoning allows it to better utilize this structure, leading to improved performance. Moreover, the iterative nature of these RAG systems can be computationally expen-

Table 11: Dataset Statistics of MuSiQue-Full dataset.

| Dataset | # Test | # Document | # Node | # Relation | # Edge |
|---|---|---|---|---|---|
| MuSiQue-Full | 2,417 | 21,100 | 19,4817 | 45,437 | 3,024,388 |

Table 12: Evaluation of G-reasoner on MuSiQue-Full dataset.

| MuSiQue-Full | EM | F1 |
|---|---|---|
| Qwen3-Emb-8B | 29.21 | 42.04 |
| HippoRAG | 24.62 | 36.16 |
| GFM-RAG | 23.40 | 33.87 |
| G-reasoner | **33.64** | **47.89** |

sive due to multiple rounds of retrieval and LLM reasoning, whereas G-reasoner achieves efficient end-to-end reasoning in a single forward pass.

## D.2 COMPARISON ON THE FULL MUSIQUE DATASET

To further validate the effectiveness of G-reasoner in real-world scenarios with a larger and noisier document corpus, we conducted additional experiments on the full dev set of the MuSiQue dataset using an expanded corpus that includes all supporting and distractor passages. The dataset statistics are summarized in Table 11. From the results in Table 12, we can observe that with the larger corpus, the performance of previous graph-based baselines (HippoRAG, GFM-RAG) drops significantly due to the increased retrieval difficulty and are even worse than conventional embedding-based methods (Qwen3-emb-8B). In contrast, G-reasoner maintains strong performance, demonstrating its robustness and effectiveness in handling larger, more complex graphs. This validates our claim that G-reasoner is applicable to real-world scenarios where knowledge is vast and diverse. Moreover, in real-world applications, G-reasoner can be further integrated with some pre-filtering retrieval methods (e.g., dense retrieval) to first narrow down the candidate documents before graph construction, making it scalable to even larger corpora.

## D.3 REASONING EXPLANATION

In addition to achieving high accuracy in final answers, G-reasoner also excels at generating reasoning explanations, as shown in Table 13. Following Xiao et al. (2025), we evaluate each method's reasoning explanations using the reasoning score (Avg R) to measure semantic alignment and consistency with ground-truth explanations, along with the Avg AR metric to assess whether the model provides correct reasoning when it answers questions accurately.

The results in Table 13 demonstrate that G-reasoner outperforms existing methods in both Avg R and Avg AR, indicating its superior ability to generate coherent and accurate reasoning explanations, reducing the hallucination of LLMs and enhancing the interpretability of the reasoning process. The case studies of the generated reasoning explanations are presented in Table 14.

Table 13: Comparison of reasoning explanation on G-bench (CS) (Xiao et al., 2025).

| Method | Avg R | Avg AR |
|---|---|---|
| GPT-4o-mini (OpenAI, 2024) | 55.5 | 39.8 |
| BM-25 (Robertson & Walker, 1994) | 59.2 | 44.2 |
| DALK (Li et al., 2024) | 58.9 | 42.1 |
| KGP (Wang et al., 2024) | 58.7 | 43.3 |
| GraphRAG (Edge et al., 2024) | 59.4 | 43.3 |
| ToG (Sun et al., 2024) | 60.1 | 44.0 |
| G-reasoner | **60.2** | **44.7** |

Table 14: Case studies for reasoning explanation of G-reasoner on G-bench (CS) (Xiao et al., 2025).

| | |
|---|---|
| Query | An upper ontology is named for its position at the top of hierarchical graphs, where it includes general concepts, with more specific concepts placed below. |
| Answer | True |
| Ground-truth Rational | An upper ontology is a framework for general concepts placed at the top of hierarchical graphs, with more specific concepts arranged beneath it. |
| Retrieved Document | The general framework of concepts is called an upper ontology because of the convention of drawing graphs with the general concepts at the top and the more specific concepts below them. |
| Generated Rational | The statement describes the structure of an upper ontology accurately. An upper ontology is indeed characterized by its hierarchical organization, where general concepts are positioned at the top, and more specific concepts are placed below them. This aligns with the provided context, which explains that the upper ontology serves as a framework for categorizing knowledge. The use of hierarchical graphs to represent these relationships further supports the truth of the statement. **ANSWER: True** |

| | |
|---|---|
| Query | Which methods are effective for preventing SQL Injection? A: Parameterized queries B: Input sanitization C: Disabling HTTP methods D: Web Application Firewalls (WAF) |
| Answer | ABD |
| Ground-truth Rational | Parameterized queries, input sanitization, and WAFs are effective strategies to prevent SQL Injection by ensuring safe data handling and filtering malicious requests. |
| Retrieved Document | 1. Use of a prepared statement would prevent this problem because the input string would have escape characters inserted. 2. To avoid this kind of SQL injection, the application should ensure that the `orderAttribute` variable value is one of the allowed values. 3. Programmers must pass user-input strings to the database only through parameters of prepared statements; creating SQL queries by concatenating strings with user-input values is an extremely serious security risk and should never be done in any program. |
| Generated Rational | To determine the effective methods for preventing SQL Injection, we can analyze each option: A: Parameterized queries — This is a well-known and effective method for preventing SQL injection. By using parameterized queries (or prepared statements), user input is treated as data rather than executable code, which prevents attackers from injecting malicious SQL. B: Input sanitization — This method involves cleaning and validating user input to ensure it does not contain harmful characters or patterns. While it can help reduce the risk of SQL injection, it is not foolproof on its own, as it can be difficult to cover all edge cases. However, it is still a useful practice when combined with other methods C: Disabling HTTP methods — This option is not directly related to preventing SQL injection. Disabling certain HTTP methods (like PUT or DELETE) can enhance security in general but does not specifically address SQL injection vulnerabilities. D: Web Application Firewalls (WAF) — A WAF can help detect and block SQL injection attempts by filtering out malicious requests before they reach the application. While it is not a substitute for secure coding practices, it is an effective layer of defense against SQL injection attacks. Based on this analysis, the correct answers are A, B, and D. C is not relevant to SQL injection prevention. **ANSWER: ABD** |

### D.4 MODEL SCALING CASE STUDY

With the implemented mixed precision training and distributed message-passing, G-reasoner can efficiently scale to larger graphs and model sizes with more computational resources. The number of required GPUs can be empirically estimated as

$$\#\text{GPU} = \frac{(|\mathcal{V}| * d) * 2.56^{-1} * 10^{-6}}{\text{GPU Memory}}, \tag{12}$$

where $|\mathcal{V}|$ is the number of nodes in the graph, $d$ is the hidden dimension of GFM. It can be helpful to estimate the required GPUs for using G-reasoner on different graph sizes and model sizes.

We illustrate some example configurations in Figure 6. From the results, with 32 A100 GPUs (80G), G-reasoner can scale to graphs with 800k nodes and a hidden dimension of 8192, which is around 2B parameters. With more GPUs, G-reasoner can further scale to larger graphs and model sizes and achieve better performance as suggested by the neural scaling law (Luo et al., 2025).

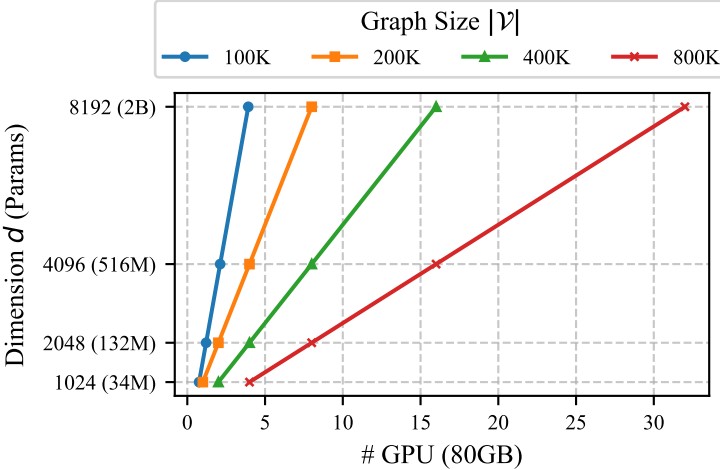

Figure 6: Scaling of G-reasoner with different model sizes and graph sizes.

### D.5 G-REASONER CASE STUDY

In this section, we first illustrate the versatile prediction results of G-reasoner. As shown in Table 15, given a query, G-reasoner can not only retrieve relevant documents to support the reasoning of LLMs, but also predict relevant entities that can be used to guide the reasoning process of LLMs. The G-reasoner exhibits great interpretability by quantifying the importance of reasoning paths. The paths' importance to the final prediction can be quantified by the partial derivative of the prediction score with respect to the triples at each layer (hop), defined as:

$$s_1, s_2, \ldots, s_L = \arg \text{top-} k \frac{\partial p_e(q)}{\partial s_l}. \tag{13}$$

The top-$k$ paths are selected based on the product of gradient scores over triples forming the path, which approximates the contribution of that path to the final prediction via the chain rule. This allows us to identify influential multi-hop reasoning chains and interpret the model's behavior. We illustrate the top-2 path interpretations in the Table 16. In the first example, the GFM identifies the path from the film entity to the director entity through the "created by" relation, and then links to the document mentioning the director. In the second example, it traces from Lady Dorothy Macmillan to her father through the "is the daughter of" relation, and then to the document mentioning him. These paths illustrate how the GFM leverages graph structure to connect entities and documents, providing interpretable reasoning chains that lead to the final answer.

Table 15: Case studies for versatile prediction of G-reasoner. Relevant predictions are highlighted in **bold**.

| Query | In which county is the town in which Raymond Robertsen was born ? |
|---|---|
| Answer | Finnmark county, |
| Supporting Documents (Title) | 1. Raymond Robertsen
2. Hammerfest |
| Entity Prediction (Top-3) | 1. cumberland county
2. **finnmark**
3. pacific county |
| Document Prediction (Top-3) | 1. **Raymond Robertsen**
2. **Hammerfest**
3. Raymond, Maine |

| Query | Who is the president of the newly declared independent country that formed the Timor Leste Commission of Truth and Friendship with the country where Pantar is found? |
|---|---|
| Answer | Francisco Guterres |
| Supporting Documents (Title) | 1. Blagar language
2. Indonesia Timor Leste Commission of Truth and Friendship
3. East Timor |
| Entity Prediction (Top-3) | 1. indonesia timor leste commission of truth and friendship
2. **francisco guterres**
3. democratic republic of timor leste |
| Document Prediction (Top-3) | 1. **Indonesia Timor Leste Commission of Truth and Friendship**
2. **East Timor**
3. **Blagar language** |

Table 16: Path interpretations of G-reasoner for multi-hop reasoning, where $r^{-1}$ denotes the inverse of the original relation, and **bold** highlights the supporting documents occurred in the paths.

| **Question** | Where was the director of *film Flags And Waves* born? |
|---|---|
| **Answer** | Toronto |
| **Supporting Docs.** | ["William Reeves (animator)", "Flags and Waves"] |
| **Paths** | 2.1465: [flags and waves (entity), is_mentioned_in, **Flags and Waves (document)**]
1.3665: [flags and waves (entity), created by, bill reeves (entity)] → [bill reeves (entity), equivalent, william reeves (entity)] → [william reeves (entity), is_mentioned_in, **William Reeves (animator) (document)** ] |
| **Question** | Where was the place of death of *Lady Dorothy Macmillan's* father? |
| **Answer** | Derbyshire |
| **Supporting Docs.** | [ "Victor Cavendish, 9th Duke of Devonshire", "Lady Dorothy Macmillan"] |
| **Paths** | 1.4286: [lady dorothy evelyn macmillan (entity), is the daughter of, victor cavendish (entity),] → [victor cavendish (entity), is_mentioned_in, **Victor Cavendish, 9th Duke of Devonshire (document)** ]
0.7685: [ lady dorothy evelyn macmillan (entity), is_mentioned_in, Lady Dorothy Macmillan (document) ] → [ **Lady Dorothy Macmillan (document)**, is_mentioned_in $^{-1}$, 9th duke of devonshire (entity) ] → [ 9th duke of devonshire (entity), holds the title of$^{-1}$, Victor Cavendish, 9th Duke of Devonshire (entity) ] → [ 9th duke of devonshire (entity), is_mentioned_in, **Victor Cavendish, 9th Duke of Devonshire (document)** ] |

## E PROMPTS

The prompts used in our experiments are presented in Figure 7. We feed the versatile predictions of G-reasoner (i.e., supporting documents and entities) to the LLMs to guide the reasoning process.

```
LLM Reasoning Prompt

As an advanced reading comprehension assistant, your task is to
analyze text passages and corresponding questions meticulously.
Your response start after "Thought: ", where you will methodically
break down the reasoning process, illustrating how you arrive at
conclusions. Conclude with "Answer: " to present a concise,
definitive response, devoid of additional elaborations.'

### Document:
<Document 1>
<Document 2>
...
<Document n>

### Entity:
<Entity 1>
<Entity 2>
...
<Entity m>

### Question:
<Question>
Thought:
```

Figure 7: The prompt template for LLM Reasoning .

## F LIMITATIONS AND FUTURE WORK

The limitations of G-reasoner are as follows: (1) The current framework is single-modality focused on text-based graphs. However, real-world knowledge often contains multi-modal data (e.g., images, audio). Extending G-reasoner to handle multi-modal graphs is an important future direction. (2) The GFM and LLMs used are integrated as separate modules. Despite the flexibility, tighter end-to-end integration may yield further performance gains, where the GFM and LLM can be co-trained to better identify and utilize graph-structured knowledge to support reasoning. (3) The G-reasoner currently focuses on question answering tasks. Extending it to other reasoning tasks (e.g., agent planning) is an interesting direction for future work.

