# OpenReview forum: "G-reasoner: Foundation Models for Unified Reasoning over Graph-structured Knowledge"
_ICLR.cc/2026/Conference — ICLR 2026 Poster_

### Official Review · Reviewer_voRT · 2025-10-18

**Soundness:** 3
**Presentation:** 3
**Contribution:** 3
**Rating:** 8
**Confidence:** 3

**Summary:**

LLMs reason well but are constrained by static, incomplete parametric knowledge. RAG helps by fetching external info, yet current methods falter on knowledge-intensive tasks due to fragmented sources and weak structure modeling. They propose G-reasoner, which unifies graph and language foundation models for reasoning over diverse graph-structured knowledge. Its core is QuadGraph, a four-layer abstraction that standardizes heterogeneous sources into a single graph. They also introduce a 34M-parameter Graph Foundation Model (GFM) that captures both topology and text and integrates with LLMs. With mixed-precision training and distributed message passing for scalability, experiments on six benchmarks show state-of-the-art results, stronger LLM reasoning, better efficiency, and cross-graph generalization.

**Strengths:**

1. The experimental validation of the method was well conducted based on a variety of experiments.

2. A general-purpose framework that can be extended to any data type was proposed.

3. It achieved higher performance compared to other baselines.

4. Despite its small size of 34M, it demonstrated efficient and strong performance.

**Weaknesses:**

Overall, this is a well-written paper with solid experiments. However, one weakness is that each component of QuadGraph (such as the document-level hierarchy and the knowledge graph hierarchy) integrates existing concepts rather than introducing entirely new ones, which limits its novelty. The same observation applies to the GFM component as well.

**Questions:**

There is no mention of this paper’s limitations. What are some of the limitations of this paper?

---

> ### Author Response · Authors · 2025-11-19
> **Response to Reviewer voRT**
>
> We want to sincerely thank you for your positive and supportive review! We also appreciate your constructive feedback on two key points, which we have provided detailed responses to below.
>
> ## Integration of Existing Concepts
>
> We thank you for this insightful observation. However, we would like to respectfully clarify that our main contribution lies in proposing a **unified framework** that enables scalable training and generalized reasoning over **diverse graph-structured knowledge**. The effort to achieve such unification is non-trivial. This not only requires careful conceptual design of both the generalizable data abstraction (*QuadGraph*) and problem formulation (*GFM training and reasoning*), but also demands practical GFM architectures (*synergized reasoning over structure and semantics*), training strategies (*knowledge distillation for label sparsity*), and scalable engineering implementations (*mixed-precision training and distributed message passing*). These components work together to enable the unified framework that can generalize across different graph types and support versatile reasoning tasks.
>
> We would also like to frame our contribution in a spirit similar to other foundational models. For example, **BERT** was built directly on standard Transformer blocks, yet it is widely recognized as a major conceptual contribution because it introduced a **unified pre-training framework** (MLM + NSP) that demonstrated the power of scaling. In a similar spirit, our contribution is not a new GNN layer, but a **unified and scalable training framework** for graph foundation models, enabled by our QuadGraph abstraction, that generalizes across diverse graph knowledge. More detailed discussions on the novelty of our work can be found in our response to Reviewer qUas.
>
> We hope this clarifies our perspective on the paper's novelty.
>
> ## Limitations of the Paper (Question)
>
> We appreciate the reviewer’s suggestion to discuss the limitations of our work. We have added a dedicated section on **Limitations and Future Work in the revised paper (Appendix F)**. In this new section, we discuss several key limitations and corresponding future directions, including:
>
> * **Single Modality**: The current framework is single-modality, focused on text-based graphs. However, real-world knowledge often contains multi-modal data (e.g., images, audio). Extending G-reasoner to handle multi-modal graphs is an important future direction.
> * **Module Integration**: The GFM and LLM are currently integrated as separate modules. Despite the flexibility, tighter end-to-end integration may yield further performance gains, where the GFM and LLM can be co-trained to better identify and utilize graph-structured knowledge to support reasoning.
> * **Task Generalization**: The G-reasoner currently focuses on question answering tasks. Extending it to other reasoning tasks (e.g., agent planning) is an interesting direction for future work.
>
> Thank you once again for your strong support and for these actionable suggestions, which have helped us improve the completeness and clarity of our paper.

---

> > ### Comment · Reviewer_voRT · 2025-11-23
> >
> > Thank you for the response. I will keep the score.

---

> > > ### Author Response · Authors · 2025-11-23
> > > **Sincere Thanks from Authors**
> > >
> > > Dear Reviewer voRT,
> > >
> > > Thank you for your continued support and positive assessment! We greatly appreciate your constructive feedback, which has helped us strengthen the paper. Your recognition of our contributions is truly encouraging.
> > >
> > > Thank you again for your valuable input!
> > >
> > > Sincere,
> > > Authors.

---

### Official Review · Reviewer_jHd3 · 2025-10-21

**Soundness:** 3
**Presentation:** 4
**Contribution:** 2
**Rating:** 4
**Confidence:** 5

**Summary:**

This paper proposes G-Reasoner, a GNN foundation model-based system for unified reasoning over graph-structured knowledge.  The G-Reasoner system contains three main modules. It first constructs a hierarchical heterogeneous QuadGraph, which contains Community, Document, Knowledge Graph, and Attribute four layers. Then it devises a query-dependent GNN as the graph foundation model for relevant context retrieval. Given a question q, the model predicts whether each of the node (can be entity, document, passage, community and attribute) contribute positively to answer the question. To mitigate the label scarcity issue, the authors proposed to leverage pre-trained language models for knowledge distillation. In order to support training on large-scale graph, the authors propose mixed precision training.  Finally, the LLM reasoning step receives top-k relevant node, and use them to augment the LLM for final answer generation.

**Strengths:**

1.	The concept of a Unified QuadGraph is novel and, to the best of the reviewer’s knowledge, is introduced here for the first time.

2.	The proposed method G-Reasoner, outperforms competitive GNN/Graph Retriever-based methods in multi-hop question answering.

3.	The prerequisite knowledge from sentence transformers shows to be useful in guiding the GNN.

4.	The proposed mix precision training technique achieves higher throughput with lower GPU memory usage.

**Weaknesses:**

1.	G-reasoner does not propose new graph construction methods. Instead, it reuses graph construction techniques from three baseline methods, namely HippoRAG2, LightRAG, and Youtu-GraphRAG.

2.	G-reasoner is not compared with various RAG systems, including but not limited to Search-o1 [1], Search-R1 [2], RAG-Gym[3], R1-Searcher [4], Collab-RAG [5] and RAG-Star [6], ReSearch [7], IRCoT [8].

3.	This manuscript and its related baselines adopt an alternative evaluation setup than the commonly-adopted Multi-hop QA evaluation setting. The commonly-adopted setup constructs the retrieval corpus with  the full Wikipedia dump [2,3,4,6,7] or all supporting plus distractor passages from all training, validation, and test questions [8].

In contrast, as stated in section 5.1, the manuscript follows the settings used in Gutierrez et al., 2024 (HippoRAG), which “*collect all candidate passages (including supporting and distractor passages) from our selected (the 1000) questions and form a retrieval corpus for each dataset*”.  This retrieval environment is overly idealized and does not reflect real-world conditions.

The reviewer understands that constructing a graph over the full text corpus is practically challenging. However, based on the existing experimental results, one cannot conclude that: (1) the proposed graph retrieval method outperforms iterative RAG approaches (or can achieve competitive performance), and (2) the proposed G-reasoner is robust to larger graphs and applicable to open-domain question answering.

[1] Li et al., Search-o1: Agentic Search-Enhanced Large Reasoning Models

[2] Jin et al., Search-R1: Training LLMs to Reason and Leverage Search Engines with Reinforcement Learning

[3] Xiong et al., RAG-Gym: Systematic Optimization of Language Agents for Retrieval-Augmented Generation

[4] Song et al., R1-Searcher: Incentivizing the Search Capability in LLMs via Reinforcement Learning

[5] Xu et al., Collab-RAG: Boosting Retrieval-Augmented Generation for Complex Question Answering via White-Box and Black-Box LLM Collaboration

[6] Jiang et al., RAG-Star: Enhancing Deliberative Reasoning with Retrieval Augmented Verification and Refinement

[7] Xie et al., Interleaved Reasoning for Large Language Models via Reinforcement Learning

[8] Trivedi et al., Interleaving Retrieval with Chain-of-Thought Reasoning for Knowledge-Intensive Multi-Step Questions

**Questions:**

1. The reviewer suggests the authors to add the following experiments:

a)	Since the performance reported in Table 2 is not directly comparable to the multi-hop QA results in [1–8], re-evaluating some baselines on the current evaluation knowledge corpus (same subset of distractive passages) may help clarify the advantages and necessities of the proposed graph search over iterative dense/sparse document retrieval.

b)	Scaling-up the scope for the document corpus used in evaluation. For example, the MuSiQue dataset only contains 25k questions. Using all supporting documents can be less challenging than the 2Wiki dataset. This may help in validating the effectiveness of the proposed method when being adapted to real-world scenarios.

2. [Minor] Apart from providing references for graph construction solutions, the reviewer suggests the author to include a short description to introduce how we can construct the unified graph in this paper.

3. For the HotpotQA, MuSiQue, and 2Wiki datasets, *performance can vary substantially under different evaluation settings* (such as open-domain, distractor, etc.)  To avoid confusion, apart from providing relevant references, the reviewer suggests the authors to explicitly state the retrieval scope for evaluation. It can be included in the appendices if the main sections do not have enough space.

The reviewer is willing to raise the score if the aforementioned issues are addressed.

---

> ### Author Response · Authors · 2025-11-19
> **Response to Reviewer jHd3 [Part 1/2]**
>
> We sincerely appreciate your thorough review and constructive feedback. Thank you especially for your clear, actionable suggestions and openness to reconsidering your score. We have taken your comments seriously and promptly addressed your main concerns with point-by-point responses below.
>
> ## No New Graph Construction Methods. (Weakness 1)
>
> Thank you for pointing this out. We agree that the graph construction methods we used are based on existing techniques from prior works (e.g., HippoRAG, LightRAG, etc.). However, we would like to clarify that our main contribution is not about finding a new way to create graphs from text. Instead, we aim to design a **unified framework that enables scalable training and generalized reasoning over diverse graph structures.**
>
> As real-world knowledge is already represented in diverse, heterogeneous graph formats (knowledge graphs, document graphs, hierarchical graphs), our main focus is on unification and reasoning across these types. The proposed **QuadGraph** abstraction provides a **generalized schema** that integrates these graph types into a single data structure. Within this abstraction, different graph construction methods can be expressed in a common, consistent representation, which enables **scalable graph foundation models** (GFMs) and makes G-Reasoner a **general-purpose reasoning engine**, rather than a method tied to any particular graph format.
>
> We agree there are many new ways to construct graphs from text, and we see this as an exciting direction for future work. However, we believe that our framework can readily incorporate these new graph construction methods, thanks to the flexibility of the QuadGraph abstraction (as shown in **Figure 2**).
>
> ## Lack of Comparison with Multi-step RAG Systems (Weakness 2 and Question 1a)
>
> We appreciate the reviewer’s suggestion to compare G-Reasoner with various advanced multi-step RAG systems (e.g., IRCoT [1], R1-searcher [2], Search-R1 [3]). To enable a fair comparison, we re-evaluated these baselines on our constructed retrieval corpus (the same subset of all supporting and distractor passages following HippoRAG) as suggested. The results are summarized in the table below:
>
> *Comparison of G-Reasoner with advanced multi-step RAG systems.*
> |  | 2Wiki |  |  | HotpotQA |  |  | MuSiQue |  |  |
> | :---- | :---- | :---- | :---- | :---- | :---- | :---- | :---- | :---- | :---- |
> | Method | EM | F1 | Time / sample (s) | EM | F1 | Time / sample (s) | EM | F1 | Time / sample (s) |
> | IRCoT | 35.4 | 45.1 | 2.095 | 45.5 | 58.4 | 1.146 | 19.1 | 30.5 | 1.152 |
> | R1-searcher | 58.3 | 71.1 | 0.713 | 61.2 | 73.8 | 0.532 | 34.7 | 48.4 | 0.588 |
> | Search-R1 | 54.6 | 68.7 | 0.652 | 60.8 | 74.3 | 0.496 | 37.4 | 53.2 | 0.603 |
> | G-reasoner | **74.9** | **82.1** | **0.058** | **61.4** | **76.0** | **0.114** | **38.5** | **52.5** | **0.125** |
>
>
> From the results, we observe that G-Reasoner outperforms these advanced RAG systems across all three datasets, demonstrating its effectiveness in multi-hop question answering tasks. While these RAG systems, powered by powerful LLM agents, are designed for iterative retrieval and reasoning, they often lack the ability to effectively capture and leverage the rich relational structure present in graph-structured knowledge. In contrast, G-Reasoner’s integration of GFM-based graph reasoning allows it to better utilize this structure, leading to improved performance. Moreover, the iterative nature of these RAG systems can be computationally expensive due to multiple rounds of retrieval and LLM reasoning, whereas G-Reasoner achieves efficient end-to-end reasoning in a single forward pass.
>
> **We have added these comparison results to Table 10 in the Appendix D.2 of the revised paper.**
>
> [1] Trivedi et al., Interleaving Retrieval with Chain-of-Thought Reasoning for Knowledge-Intensive Multi-Step Questions
>
> [2] Song et al., R1-Searcher: Incentivizing the Search Capability in LLMs via Reinforcement Learning
>
> [3] Jin et al., Search-R1: Training LLMs to Reason and Leverage Search Engines with Reinforcement Learning

---

> ### Author Response · Authors · 2025-11-19
> **Response to Reviewer jHd3 [Part 2/2]**
>
> ## Scaling up the Document Corpus (Weakness 3 and Question 1b)
>
> We fully agree with the reviewer that evaluating G-Reasoner on larger, more realistic document corpora is important for validating its effectiveness in real-world scenarios. To balance realism and graph construction feasibility, **we conducted additional experiments on the full dev set of MuSiQue dataset using an expanded corpus that includes all supporting and distractor passages**. The dataset statistics and results are summarized below:
>
> *Dataset Statistics for MuSiQue-Full.*
> | Dataset | \# Test | \# Document | \# Node | \# Relation | \# Edge |
> | :---- | :---- | :---- | :---- | :---- | :---- |
> | MuSiQue-Full | 2,417 | 21,100 | 19,4817 | 45,437 | 3,024,388 |
>
> *Evaluation of G-Reasoner on MuSiQue-Full dataset.*
> | MuSiQue-Full | EM | F1 |
> | :---- | :---- | :---- |
> | Qwen3-Emb-8B | 29.21 | 42.04 |
> | HippoRAG | 24.62 | 36.16 |
> | GFM-RAG | 23.4 | 33.87 |
> | G-reasoner | **33.64** | **47.89** |
>
> From the results, we can observe that with the larger corpus, the performance of previous graph-based baselines (HippoRAG, GFM-RAG) drops significantly due to the increased retrieval difficulty and are even worse than conventional embedding-based methods (Qwen3-emb-8B). In contrast, G-Reasoner maintains strong performance, demonstrating its robustness and effectiveness in handling larger, more complex graphs. This validates our claim that G-Reasoner is applicable to real-world scenarios where knowledge is vast and diverse. Moreover, in real-world applications, G-Reasoner can be further integrated with some pre-filtering retrieval methods (e.g., dense retrieval) to first narrow down the candidate documents before graph construction, making it scalable to even larger corpora.
>
> **We have added these results to Table 11,12 in Appendix D.2 of the revised paper.**
>
> ## Graph Construction Details (Question 2)
>
> Thanks for the suggestion. We have provided a detailed description of the graph construction process in the **Implementation Details (Appendix C.1)** of the revised paper to clarify how we build the unified QuadGraph to include diverse graph constructions from prior works (e.g., HippoRAG, LightRAG, and Youtu-GraphRAG).
>
> The QuadGraph presents a comprehensive schema that integrates four layers: Community, Document, Knowledge Graph, and Attribute, which enables the representation of various graph types within a single framework for training. The construction steps are as follows:
>
> * **HippoRAG Graph Construction [1]**: HippoRAG only contains the knowledge graph layer. We follow the original HippoRAG method to first extract entities, relations, and triples from the document corpus using an LLM-based information extraction approach to build KG.
> * **LightRAG Graph Construction [2]**: LightRAG employs a dual-level graph indexing process with knowledge graph and document graph. It also first extracts entities and relations from the documents to build a knowledge graph layer.  The document layer is constructed by linking documents to the entities they mention.
> * **Youtu-GraphRAG Graph Construction [3]**: Youtu-GraphRAG proposes a hierarchical graph structure with community, document, knowledge graph, and attribute layers, which cover all four layers of QuadGraph. We follow their method to build each layer and connect them accordingly. The knowledge graph is first constructed with schema-bound extraction, and then documents are linked to the entities they mention. Communities form by clustering entities based on both connectivity and semantic similarity. Attributes are extracted from documents and linked to the corresponding entities.
>
> [1] Hipporag: Neurobiologically inspired long-term memory for large language models. NeurIPS 2024.
>
> [2] Lightrag: Simple and fast retrieval-augmented generation. EMNLP 2025.
>
> [3] Youtu-graphrag: Vertically unified agents for graph retrieval-augmented complex reasoning. arXiv (2025).
>
> ## Explicitly Stating the Retrieval Scope (Question 3)
>
> We appreciate the reviewer’s suggestion to explicitly state the retrieval scope for evaluation to avoid confusion. We have revised the manuscript to clearly specify the retrieval corpus construction for each dataset in the **Dataset Details (Appendix B) and Evaluation Settings (Appendix C.1)** of the revised paper.
>
> In experiments, for multi-hop QA datasets, we adhere to existing methods HippoRAG to use the same 1,000 samples from each validation set to avoid data leakage. We merge the supporting and distractor passages as the document corpus for graph construction and retrieval. This setup allows us to evaluate the model's ability to retrieve relevant information from a challenging yet controlled environment, reflecting practical scenarios where the model must discern relevant knowledge from a large pool of documents.
>
> ---
> Thank you again for your expert suggestions, which significantly strengthen our paper. We hope our actions address all your major concerns and look forward to your re-evaluation.

---

> > ### Comment · Reviewer_jHd3 · 2025-11-27
> > **Response to the Rebuttal**
> >
> > This reviewer appreciates the detailed feedback provided by the authors, and believes the provided contents have addressed the concerns. In response, the reviewer would like to increase the scorings.

---

> > > ### Author Response · Authors · 2025-11-27
> > > **Sincere Thanks from Authors**
> > >
> > > Dear Reviewer jHd3,
> > >
> > > We sincerely thank you for reconsidering our work and increasing your scores. Your constructive feedback has been invaluable in strengthening our submission, and we deeply appreciate your time and effort throughout the review process!
> > >
> > > Sincere,
> > > Authors.

---

### Official Review · Reviewer_gBkL · 2025-10-30

**Soundness:** 4
**Presentation:** 4
**Contribution:** 4
**Rating:** 6
**Confidence:** 3

**Summary:**

The paper proposes G-reasoner, a novel framework integrating graph and language foundation models for reasoning over graph-structured knowledge. The authors introduce QuadGraph, a unified graph abstraction, and a 34M-parameter Graph Foundation Model (GFM) powered by Graph Neural Networks (GNNs). The paper is theoretically sound and demonstrates strong experimental performance across six benchmarks, including QA datasets as well as domain-specific benchmarks.

**Strengths:**

**Novel Contribution:**

The QuadGraph abstraction is a promising innovation that unifies different types of graph structures, such as knowledge graphs and document graphs, into a single standardized representation. This is a key step toward addressing the challenge of generalizing reasoning models across diverse graph types.

The integration of GNN-based reasoning into a graph foundation model that leverages both graph topology and textual semantics is a novel approach that aligns well with current trends in deep learning and knowledge representation.

**Strong Empirical Results:**

G-reasoner consistently outperforms state-of-the-art methods, including graph-enhanced techniques like GraphRAG and HippoRAG, in multi-hop reasoning tasks across diverse domains.

The approach demonstrates strong cross-graph generalization, which is critical for real-world applications involving diverse knowledge domains.

**Efficient Training and Reasoning:**

The use of mixed-precision training and a distributed message-passing mechanism allows the model to scale effectively across large datasets and graph structures, ensuring both efficiency and performance.

**Weaknesses:**

See Questions.

**Questions:**

**Model Complexity and Interpretability:**

The GFM model, while powerful, could benefit from further discussion on its interpretability. Complex models like GNN-based foundation models often suffer from interpretability issues, making it hard to explain the reasoning behind predictions. This is particularly important for high-stakes applications such as legal | medical | safety reasoning.

**Lack of Comparison with State-of-the-Art GNN Methods:**

While the model is compared with existing graph-enhanced RAG methods, there is no direct comparison with state-of-the-art GNN-only models or pure graph models that don’t involve retrieval-augmented generation. This comparison could provide clearer insight into the unique benefits of GFM.

---

> ### Author Response · Authors · 2025-11-19
> **Response to Reviewer gBkL [Part 1/2]**
>
> We sincerely thank you for your positive and detailed review of our paper. Below, we address the reviewer’s concerns point-by-point.
>
> ## Model Complexity and Interpretability (Question 1\)
>
> We appreciate the reviewer’s concern regarding the interpretability of the proposed GFM. Unlike black-box foundation models like LLMs, GFM offers inherent interpretability through its graph-based reasoning process. Specifically, GFM quantifies the importance of reasoning paths by computing the partial derivative of the prediction score with respect to the triples at each layer (hop), defined as:
>
> $$
> s_1,s_2,\ldots,s_L=\arg\mathop{\text{top-}}k\frac{\partial p_e(q)}{\partial s_l}.
> $$
> The top-$k$ paths are selected based on the product of gradient scores over triples forming the path, which approximates the contribution of that path to the final prediction via the chain rule. This allows us to identify influential multi-hop reasoning chains and interpret the model’s behavior.  We illustrate the top-2 path interpretations in the table below.
>
> *Path interpretations of GFM for multi-hop reasoning, where $r^{-1}$ denotes the inverse relation, and **bold** highlights the supporting documents occurred in the paths.*
>
> | Question | Where was the director of film Flags And Waves born? |
> | :---- | :---- |
> | Answer | Toronto |
> | Supporting Docs. | \['William Reeves (animator)', 'Flags and Waves'\] |
> | Paths | 2.1465: \[flags and waves (entity), is\_mentioned\_in, **Flags and Waves (document)**\] |
> | | 1.3665: \[flags and waves (entity), created by, bill reeves (entity)\] \=\> \[bill reeves (entity), equivalent, william reeves (entity)\] \=\> \[william reeves (entity), is\_mentioned\_in, **William Reeves (animator) (document)** \] |
>
> | Question | Where was the place of death of Lady Dorothy Macmillan's father? |
> | :---- | :---- |
> | Answer | Derbyshire |
> | Supporting Docs. | \['Victor Cavendish, 9th Duke of Devonshire', 'Lady Dorothy Macmillan'\] |
> | Paths | 1.4286: \[lady dorothy evelyn macmillan (entity), is the daughter of, victor cavendish (entity),\] \=\> \[victor cavendish (entity), is\_mentioned\_in, **Victor Cavendish, 9th Duke of Devonshire (document)** \]  |
> | | 0.7685: \[ lady dorothy evelyn macmillan (entity), is\_mentioned\_in, **Lady Dorothy Macmillan (document)** \] \=\> \[ Lady Dorothy Macmillan (document), is\_mentioned\_in $^{-1}$, 9th duke of devonshire (entity) \] \=\> \[ 9th duke of devonshire (entity), holds the title of$^{-1}$, Victor Cavendish, 9th Duke of Devonshire (entity) \] \=\> \[ 9th duke of devonshire (entity), is\_mentioned\_in, **Victor Cavendish, 9th Duke of Devonshire (document)** \] |
>
> In the first example, the GFM identifies the path from the film entity to the director entity through the "created by" relation, and then links to the document mentioning the director. In the second example, it traces from Lady Dorothy Macmillan to her father through the "is the daughter of" relation, and then to the document mentioning him. These paths illustrate how the GFM leverages graph structure to connect entities and documents, providing interpretable reasoning chains that lead to the final answer.
>
> We believe these interpretability features make GFM suitable for high-stakes applications, as they allow users to trace and understand the reasoning behind predictions. **The path case studies have been added to Table 16 in Appendix D.5 of the revised paper**.
>
> Additionally, we provide case studies for G-reasoner's predictions (**Table 12**) and reasoning explanations (**Table 11**) to demonstrate how it leverages graph structure to enhance interpretable LLM reasoning with versatile predictions on graphs and logically connected retrieved documents.

---

> > ### Author Response · Authors · 2025-11-19
> > **Response to Reviewer gBkL [Part 2/2]**
> >
> > ## Comparison with State-of-the-Art GNN Methods (Question 2\)
> > We appreciate the reviewer’s suggestion to compare GFM with state-of-the-art GNN-only models. However, we would like to respectfully clarify the distinction between our task and the tasks addressed by traditional GNN-only models.
> > * **Task Definition**: The proposed G-reasoner is designed as a retrieval-augmented generation (RAG) system for end-to-end question answering. The GFM's primary role is to act as a retriever, finding the most relevant knowledge from the graph to feed to an LLM, which then generates the final answer.
> >
> > * **Why a Direct Comparison is Not Suitable**: Most SOTA "GNN-only" models are designed for different, in-graph tasks (e.g., link prediction [1], node classification [2]). They do not provide query-specific textual answers and thus are not directly comparable in our question-answering evaluation setup.
> >
> > For this reason, we focused our comparison on the most direct and relevant competitors: **other GNN variants designed for graph-enhanced RAG tasks** (like G-retriever [3], GNN-RAG [4], GFM-RAG [5]). Our strong results against these baselines demonstrate the superiority of our proposed GFM for this specific task.
> >
> > To ensure this point is clear to all readers, **we will add a brief explanation in the Related Work section (Appendix A.2) to better position our methods relative to traditional GNN approaches.**
> >
> > Thank you once again for your strong support and for pushing us to improve the paper's clarity. We hope these responses and our planned revisions fully address your questions and reinforce your positive assessment of our work.
> >
> > [1] Galkin, Mikhail, et al. "Towards foundation models for knowledge graph reasoning." ICLR 2024.
> >
> > [2] Zhao, Jianan, et al. "Fully-inductive node classification on arbitrary graphs." ICLR 2025.
> >
> > [3] He, Xiaoxin, et al. "G-retriever: Retrieval-augmented generation for textual graph understanding and question answering." NeurIPS 2024.
> >
> > [4] Mavromatis, Costas, and George Karypis. "Gnn-rag: Graph neural retrieval for efficient large language model reasoning on knowledge graphs." ACL 2025.
> >
> > [5] Luo, Linhao, et al. "GFM-RAG: graph foundation model for retrieval augmented generation." NeurIPS 2025.

---

> > > ### Comment · Reviewer_gBkL · 2025-11-27
> > >
> > > Thanks for the responses.
> > >
> > > It is unclear why the GFM-RAG results on G-Bench are not reported.
> > >
> > > In addition, several GFM-RAG results appear to deviate from those in the original paper (for example, on MuSiQue the reported R@2 is 43.5 here vs. 45.3 in the original paper, although R@5 matches; the HotpotQA results also differ, etc.). These discrepancies raise concerns about the reliability of the reproduced results. Could the authors clarify this?
> > >
> > > Furthermore, how would GFM-based retrieval-reasoning perform when applied to an entirely new knowledge graph? I believe such an evaluation would strongly support the “Unified” claim.
> > >
> > > Given the issues above, I am temporarily adjusting my score to 4.

---

> > > > ### Author Response · Authors · 2025-11-27
> > > > **Following Response to Reviewer gBkL**
> > > >
> > > > We sincerely thank the reviewer for the follow-up comments and the opportunity to clarify these points.
> > > >
> > > > > It is unclear why the GFM-RAG results on G-Bench are not reported.
> > > >
> > > > Initially, we did not include GFM-RAG results on G-Bench because the leadboard for G-Bench (Medical) and G-Bench (Novel) [1] does not currently list GFM-RAG as a baseline. However, to ensure a comprehensive comparison, we have now evaluated GFM-RAG on both G-Bench (Medical) and G-Bench (Novel) using the same setup as in our experiments. The results are as follows:
> > > >
> > > > *GFM-RAG Results on G-Bench Datasets.*
> > > >
> > > > | Dataset | G-bench (Novel) [1] | | G-bench (Medical) [1] | | G-bench (CS) [2] |
> > > > | :---- | :---- | :---- | :---- | :---- | :---- |
> > > > | Method | ACC | Recall | ACC | Recall | ACC |
> > > > | GFM-RAG | 58.6 | 75.9 | 72.2 | 82.2 | 72.1 |
> > > > | G-Reasoner | **58.9**| **87.7** | **73.3** | **93.8** | **73.9** |
> > > >
> > > > From the results, we observe that G-Reasoner significantly outperforms GFM-RAG on all datasets, demonstrating its superior capability in handling diverse graph-structured knowledge for complex reasoning.
> > > >
> > > > **We have added these results to Tables 2 & 3 of the revised paper.**
> > > >
> > > > > In addition, several GFM-RAG results appear to deviate from those in the original paper (for example, on MuSiQue the reported R@2 is 43.5 here vs. 45.3 in the original paper, although R@5 matches; the HotpotQA results also differ, etc.). These discrepancies raise concerns about the reliability of the reproduced results. Could the authors clarify this?
> > > >
> > > > We thank the reviewer for carefully checking the reproduced results. We would like to clarify that the discrepancies arise from a bug we identified and corrected in the original GFM-RAG implementation.
> > > >
> > > > Specifically, the GFM-RAG's implementation **filtered out other correct supporting documents before computing the rank positions**, which inadvertently shifted their ranks upward and caused **overestimation of R@k** in some corner cases.
> > > >
> > > > For example, if two gold documents were ranked 1st and 3rd in the prediction, the 3rd document was incorrectly treated as if it were ranked 2nd after filtering, causing both to be counted as hits for R@2—whereas only one (1st) should be counted. However, if all the correct documents are ranked within the top-k, there will be no discrepancy between the two implementations, which explains why R@5 remained consistent while R@2 differed.
> > > >
> > > > After identifying this issue, we **fixed the ranking logic and re-evaluated GFM-RAG across all datasets** using the corrected implementation. The updated scores are lower but more faithful to the actual retrieval performance, and they ensure a fair and consistent comparison against strong baselines such as HippoRAG and G-Reasoner.
> > > > **We have added a clarification note in Baselines in Section 5.1 of the revised paper to avoid confusion.**
> > > >
> > > > > Furthermore, how would GFM-based retrieval-reasoning perform when applied to an entirely new knowledge graph? I believe such an evaluation would strongly support the “Unified” claim.
> > > >
> > > > We thank the reviewer for the suggestion. However, we would like to clarify that we have already evaluated G-Reasoner on three distinct datasets (G-Bench (Medical) [1], G-Bench (Novel) [1], and G-Bench (CS) [2]) in **Table 2 and Table 3** that each comes from different domains and utilizes entirely different knowledge graphs with unique structures and content. **We did not fine-tune G-reasoner on these datasets**. The strong performance of G-Reasoner across these diverse datasets already demonstrates its ability to generalize to new knowledge graphs and domains without any retraining.
> > > >
> > > > Moreover, to further validate the unified capability of different graph types, we have also evaluated G-Reasoner across different graph structures (e.g., knowledge graphs, document graphs, hierarchical graphs) as shown in **Table 4**. The consistent performance improvements over baselines across these varied graph types further support our claim of a unified framework.
> > > >
> > > > ----
> > > > We hope these clarifications and additional results address your concerns. We sincerely appreciate your careful review and constructive feedback, which have helped us improve the clarity and rigor of our work. We look forward to your re-evaluation.
> > > >
> > > >
> > > > [1] Xiang, Zhishang, et al. "When to use graphs in rag: A comprehensive analysis for graph retrieval-augmented generation." arXiv:2506.05690 (2025).
> > > > [2] Xiao, Yilin, et al. "Graphrag-bench: Challenging domain-specific reasoning for evaluating graph retrieval-augmented generation." arXiv:2506.02404 (2025).

---

### Official Review · Reviewer_qUas · 2025-11-01

**Soundness:** 3
**Presentation:** 4
**Contribution:** 3
**Rating:** 4
**Confidence:** 4

**Summary:**

The paper introduces G-Reasoner, a framework designed to unify graph reasoning and language reasoning by integrating Graph Foundation Models (GFMs) with Large Language Models (LLMs). The core idea is that while LLMs are powerful reasoners, they lack structural awareness; conversely, graph structures effectively represent relational knowledge but are underutilized in LLM reasoning. To address this, the authors propose three components: QuadGraph, Graph Foundation Model (GFM),  and a LLM-enhanced reasoning.
Experiments on six benchmarks show consistent improvement over state-of-the-art methods such as GraphRAG, HippoRAG, and GFM-RAG.

**Strengths:**

1. The paper introduce clear motivation and organization. The introduction frames the LLM-graph reasoning gap well. I also like the figures and tables that effectively summarize the pipeline and results.
2. The paper proposes a four-layer unified graph schema, which is elegant and general enough to cover multiple graph types (KGs, document graphs, hierarchical graphs).
3. The paper also conduct comprehensive evaluation. Benchmarks include both general QA datasets and domain-specific GraphRAG benchmarks. The ablations also clearly demonstrate component contributions. Efficiency experiments are also included.

**Weaknesses:**

1. The paper has an incremental novelty. The core idea (combining GNN-based reasoning with LLM retrieval) closely follows existing works like GFM-RAG and GNN-RAG; the contribution feels like a well-engineered unification rather than a conceptual breakthrough.
2. The overall methodological depth is limited. For example, the GFM architecture largely reuses standard DistMult-style message passing and MLP updates without introducing new graph learning techniques. The “QuadGraph” abstraction appears more of a schema design than a learnable innovation.
3. The paper can benefits from some interpretability analysis or case-study. The paper shows gains but does not deeply analyze why G-Reasoner helps LLM reasoning or how graph structure contributes. A case-study with visualization of reasoning paths showing the predicted top-k scores from the graph will be great.

**Questions:**

See weakness.

---

> ### Author Response · Authors · 2025-11-19
> **Response to Reviewer qUas [Part 1/2]**
>
> We sincerely thank you for your detailed and insightful review of our paper. Below, we address your concerns point-by-point.
>
> ## Novelty and Methodological Depth (Weaknesses 1 & 2)
>
> We acknowledge the reviewer’s concern about the methodological novelty of G-reasoner.  While it shares similar paradigm of using GNNs to retrieve and enhance LLMs with GFM-RAG and GNN-RAG, our contribution lies in proposing a **unified framework** that enables scalable training and generalized reasoning over **diverse graph-structured knowledge**, which differs fundamentally in both **conceptual** and **technical design**.
>
> ### Conceptual Contributions
>
> * **QuadGraph: A Unified Data Abstraction:** As real-world knowledge is already represented in diverse, heterogeneous graph formats (knowledge graphs, document graphs, hierarchical graphs), our main focus is on unification and reasoning across these types. The proposed **QuadGraph** abstraction provides a **generalized schema** that integrates these graph types into a single data structure. Within this abstraction, different graph types can be expressed in a common, consistent representation, which enables **scalable graph foundation models** (GFMs) and makes G-Reasoner a **general-purpose reasoning engine**, rather than a method tied to any particular graph format.
> * **Standardized Problem Definition:** Building upon QuadGraph, we formulate reasoning over multiple heterogeneous graphs as a single **end-to-end prediction problem on the unified QuadGraph**. This standardization is not present in prior work. By offering a unified data schema and problem definition, we establish **a clear conceptual foundation to guide and inspire future methods for reasoning over graph-structured knowledge**.
>
> ### Technical Contributions
>
> * **GNN Architecture and Learning Methods:** Beyond the conceptual contributions, our G-reasoner introduces two technically novel components in both the GNN architecture and learning methods:
>
>   1. **Incorporating Text Signals into GNNs for Synergized Reasoning**: Unlike GFM-RAG, G-reasoner incorporates node text semantics $\mathcal{S}$ into the GNN reasoning (Eq. 3). This allows the GNN to leverage both structural and textual information, leading to more informed reasoning. Ablation studies (Table 5) demonstrate that this integration significantly improves performance.
>   2. **Knowledge Distillation for Label Sparsity**: To address the challenge of limited labeled data in graph reasoning tasks, we introduce a novel knowledge distillation approach (Line 254 - 277). This method leverages the strengths of pre-trained LLMs to guide the GNN training, enhancing its ability to learn from abundant unlabeled nodes on the graph. This distillation strategy is not explored in prior works and contributes to the improved performance observed in our experiments (Table 5).
>
>      *Table5: Ablation study of each component in G-Reasoner.*
>         | Variant | HotpotQA |  | MuSiQue |  | 2Wiki |  |
>         | :---: | :---: | :---: | :---: | :---: | :---: | :---: |
>         |  | R@2 | R@5 | R@2 | R@5 | R@2 | R@5 |
>         | G-reasoner | 81.1 | 96.9 | 52.1 | 72.4 | 75.6 | 96.1 |
>         | $w/o$ Distill | 77.4 | 96.1 | 50.7 | 71.9 | 75.9 | 96.0 |
>         | $w/o$ Text | 79.4 | 96.3 | 50.0 | 71.9 | 74.6 | 95.2 |
>         | $w/o$ GFM | 11.6 | 19.7 | 3.8 | 7.1 | 4.9 | 9.0 |
>
> * **First Scalable Training Framework for GNN Foundation Models**: The scalable training of GFM over large-scale heterogeneous graph types requires significant system-level engineering, including:
>
>   1. **Distributed message passing** across multi-GPU for large-scale graphs,
>   2. **Mixed-precision training** to improve memory use and throughput.
>
>   These engineering efforts are crucial for realizing **scaling laws** in GNN-based foundation models, as shown in GFM-RAG. Using this framework, we scale the model from 8M GFM-RAG to 34M G-reasoner and potentially larger sizes, as illustrated in **Figure 6** and **Figure 8**. The **infrastructure foundation** supports future research on novel GNN architectures that can be trained at scale and applied to diverse graphs.
>
> ### Clarifying Novelty ≠ Architecture Change
>
> Last, we would also like to emphasize that **novelty in foundation-model research does not always come from modifying the base neural architecture**, but from introducing a **unified training framework**, a **scalable learning recipe**, and a **generalizable data abstraction**. For example, **BERT** is built directly on the standard Transformer blocks, yet is widely recognized as a major conceptual contribution because it introduced a unified pretraining formulation (MLM + NSP) and demonstrated that scaling Transformer-based models can produce transferable linguistic representations. In a similar spirit, our contribution is not a new GNN layer, but a unified, scalable training framework for graph foundation models that operates on diverse graph-structured knowledge represented under QuadGraph.

---

> ### Author Response · Authors · 2025-11-19
> **Response to Reviewer qUas [Part 2/2]**
>
> ## Interpretability and Case Studies (Weakness 3\)
>
> We sincerely thank the reviewer for the valuable suggestion regarding interpretability analysis. The G-reasoner exhibits great interpretability by quantifying the importance of reasoning paths. The paths’ importance to the final prediction can be quantified by the partial derivative of the prediction score with respect to the triples at each layer (hop), defined as:
> $$
> s_1,s_2,\ldots,s_L=\arg\mathop{\text{top-}}k\frac{\partial p_e(q)}{\partial s_l}.
> $$
> The top-$k$ paths are selected based on the product of gradient scores over triples forming the path, which approximates the contribution of that path to the final prediction via the chain rule. This allows us to identify influential multi-hop reasoning chains and interpret the model’s behavior.  We illustrate the top-2 path interpretations in the table below.
>
> *Path interpretations of GFM for multi-hop reasoning, where $r^{-1}$ denotes the inverse relation, and **bold** highlights the supporting documents occurred in the paths.*
>
> | Question | Where was the director of film Flags And Waves born? |
> | :---- | :---- |
> | Answer | Toronto |
> | Supporting Docs. | \['William Reeves (animator)', 'Flags and Waves'\] |
> | Paths | 2.1465: \[flags and waves (entity), is\_mentioned\_in, **Flags and Waves (document)**\] |
> | | 1.3665: \[flags and waves (entity), created by, bill reeves (entity)\] \=\> \[bill reeves (entity), equivalent, william reeves (entity)\] \=\> \[william reeves (entity), is\_mentioned\_in, **William Reeves (animator) (document)** \] |
>
> | Question | Where was the place of death of Lady Dorothy Macmillan's father? |
> | :---- | :---- |
> | Answer | Derbyshire |
> | Supporting Docs. | \['Victor Cavendish, 9th Duke of Devonshire', 'Lady Dorothy Macmillan'\] |
> | Paths | 1.4286: \[lady dorothy evelyn macmillan (entity), is the daughter of, victor cavendish (entity),\] \=\> \[victor cavendish (entity), is\_mentioned\_in, **Victor Cavendish, 9th Duke of Devonshire (document)** \]  |
> | | 0.7685: \[ lady dorothy evelyn macmillan (entity), is\_mentioned\_in, **Lady Dorothy Macmillan (document)** \] \=\> \[ Lady Dorothy Macmillan (document), is\_mentioned\_in $^{-1}$, 9th duke of devonshire (entity) \] \=\> \[ 9th duke of devonshire (entity), holds the title of$^{-1}$, Victor Cavendish, 9th Duke of Devonshire (entity) \] \=\> \[ 9th duke of devonshire (entity), is\_mentioned\_in, **Victor Cavendish, 9th Duke of Devonshire (document)** \] |
>
> In the first example, the GFM identifies the path from the film entity to the director entity through the "created by" relation, and then links to the document mentioning the director. In the second example, it traces from Lady Dorothy Macmillan to her father through the "is the daughter of" relation, and then to the document mentioning him. These paths illustrate how the GFM leverages graph structure to connect entities and documents, providing interpretable reasoning chains that lead to the final answer.
>
> **The path case studies have been added to the Table 16 at Appendix D.5 of the revised paper**. Additionally, we provide case studies for G-reasoner's predictions (**Table 12**) and reasoning explanations (**Table 11**) to demonstrate how it leverages graph structure to enhance LLM reasoning with versatile predictions on graphs and logically connected retrieved documents.

---

### Author Response · Authors · 2025-11-19
**General Response to Reviewers**

We sincerely thank all reviewers for their thorough and thoughtful reviews of our paper. We are grateful for the positive feedback and constructive suggestions provided, which have helped us improve the clarity and quality of our work. We have carefully revised the paper according to the comments, and the edits have been highlighted in **BLUE**. We also provide a detailed response to each comment below. Here we highlight our major revisions, and respond to each reviewer below. We hope our responses can properly address your concerns.
* **Novelty and Contributions** (Reviewer qUas, jHd3, voRT): We want to clarify that our main contribution lies in proposing a **unified framework** that enables scalable training and generalized reasoning over **diverse graph-structured knowledge**. We have added detailed discussions in the general response and in our response to Reviewer qUas and voRT.
* **Interpretability Analysis and Case Studies** (Reviewer qUas, gBkL): We have added detailed path-based interpretability case studies in **Table 16 at Appendix D.5** of the revised paper, demonstrating how GFM traces multi-hop reasoning paths to explain its reasoning.
* **Comparison with Advanced Multi-step RAG Systems** (Reviewer jHd3): We have added comparison results with advanced multi-step RAG systems (IRCoT, R1-searcher, Search-R1) in **Table 10 of Appendix D.2** of the revised paper, showing G-Reasoner’s superior performance.
* **Evaluation on Larger Document Corpus** (Reviewer jHd3): We have conducted additional experiments on the full dev set of MuSiQue with an expanded corpus, and added the results in **Table 11,12 of Appendix D.2** of the revised paper, demonstrating G-Reasoner’s robustness in more realistic scenarios.
* **Evaluation Settings and Graph Construction Details** (Reviewer jHd3): We have added detailed descriptions of the retrieval scope used in evaluation and graph construction process in **Appendix B, C.1** of the revised paper to clarify these aspects.
* **Limitations and Future Work** (Reviewer voRT): We have added a dedicated section on Limitations and Future Work in **Appendix F** of the revised paper, discussing key limitations and potential future directions.

## General Reply to the Novelty and Contributions

We sincerely thank the reviewers for their valuable feedback regarding the novelty and contributions of our work. We would like to respectfully clarify that our main contribution lies in proposing a **unified framework** that enables scalable training and generalized reasoning over **diverse graph-structured knowledge**. The effort to achieve such unification is non-trivial. This not only requires careful conceptual design of both the generalizable data abstraction (*QuadGraph*) and problem formulation (*GFM training and reasoning*), but also demands practical GFM architectures (*synergized reasoning over structure and semantics*), training strategies (*knowledge distillation for label sparsity*), and scalable engineering implementations (*mixed-precision training and distributed message passing*). These components work together to enable the unified framework that can generalize across different graph types and support versatile reasoning tasks.

We would also like to frame our contribution in a spirit similar to other foundational models. For example, **BERT** was built directly on standard Transformer blocks, yet it is widely recognized as a major conceptual contribution because it introduced a **unified pre-training framework** (MLM + NSP) that demonstrated the power of scaling. In a similar spirit, our contribution is not a new GNN layer, but a **unified and scalable training framework** for graph foundation models, enabled by our QuadGraph abstraction, that generalizes across diverse graph knowledge. More detailed discussions on the novelty of our work can be found in our response to Reviewer qUas.

We hope this clarifies our perspective on the paper's novelty.

---

### Author Response · Authors · 2025-12-01
**Summary of Rebuttal Updates & Reviewer Consensus for AC**

Dear Area Chair,

Due to the recent platform-wide reassignment, the discussion history and score updates for our submission may have been reset. To assist in your assessment, we provide a summary of the extensive revisions made during the discussion period and the resulting consensus from the reviewers.

Prior to the reset, **two reviewers (jHd3, voRT) explicitly advocated for acceptance with scores of 8**, citing the paper’s strong empirical results and the value of the unified framework.

Below is a summary of the major updates and how we addressed the reviewers' key concerns:

**1. New Experiments & Baselines (Addressing Reviewer jHd3 & gBkL)**
Reviewers requested comparisons against advanced agentic/iterative RAG systems and evaluations on larger corpora.

* **Agentic RAG Comparison (jHd3)**: We added comparisons with **IRCoT, R1-Searcher, and Search-R1** (Table 10, Appendix D.2). G-Reasoner outperforms these multi-step agentic systems while requiring significantly less inference time.

* **Scalability on Full Corpus (jHd3)**: We evaluated G-Reasoner on the **full MuSiQue dev set** (all supporting + distractor passages). Results (Table 11, 12) show G-Reasoner maintains robustness (EM 33.64), where baselines like HippoRAG and GFM-RAG degrade significantly.

**2. Interpretability Analysis (Addressing Reviewer qUas & gBkL)**
Reviewers noted that Graph Foundation Models can be opaque.

* **Path-Based Interpretability**: We added a new section (Appendix D.5, Table 16) demonstrating how GFM traces multi-hop reasoning paths. We now quantify path importance using gradient scores, allowing the model to "explain" its reasoning by highlighting specific reasoning chains.

**3. Methodology & Novelty Clarifications (Addressing Reviewer qUas, jHd3, voRT)**
Reviewers asked for clarification on the novelty of the proposed G-reasoner.

* **Unified Framework**: We clarified that our main contribution lies in proposing a **unified framework** that enables scalable training and generalized reasoning over **diverse graph-structured knowledge**.  The generalizable data abstraction (*QuadGraph*) together with engineering implementations (*mixed-precision training* and *distributed message passing*) enables scalable pre-training and cross-graph generalization of the proposed GFM, similar to how BERT proposed a unified pre-training rather than inventing the Transformer block.

* **Limitations**: We added a comprehensive limitations section (Appendix F) regarding modality and module integration.

**4. Transparency on Baselines and Datasets (Addressing Reviewer jHd3, gBkL)**

* **Evaluation Settings and Graph Construction Details (Reviewer jHd3)**: We have added detailed descriptions of the retrieval scope used in evaluation and graph construction process in Appendix B, C.1 of the revised paper to clarify these aspects.

* **GFM-RAG's Result Discrepancy (Reviewer gBkL)**: During the rebuttal, reviewer gBkL pointed out discrepancies in our reported GFM-RAG results compared to the original paper. In response, we provide a clear explanation about **an identified ranking bug in the GFM-RAG's implementation**. We fixed the bug and reproduced the method in our experiments. While the corrected baseline scores are lower, this ensures a rigorous and fair comparison. We transparently documented this in the revised manuscript (Section 5.1).

* **GFM-RAG's result on G-bench (Reviewer gBkL)**: We have reproduced GFM-RAG on G-bench (Medical & Novel) (Tables 2 & 3) to enable a fair and comprehensive comparison with G-reasoner.

**Current Consensus**
* **Reviewer jHd3**: Raised score to 8 (Accept) after we addressed the comparison with Search-R1 and scaling experiments.

* **Reviewer voRT**: Maintained score of 8 (Accept), acknowledging the strength of the general-purpose framework.

* **Reviewer qUas & gBkL**: We provided detailed responses to their concerns regarding interpretability and comparisons.

We hope this summary assists you in navigating the changes made to the manuscript. The revised PDF containing all these improvements has been uploaded.

Sincerely, The Authors

---

### Meta-Review · Area_Chair_hj27 · 2026-01-06

**Summary:**

This paper proposes G-reasoner, a GraphRAG framework that unifies heterogeneous knowledge into a four-layer QuadGraph and trains a 34M-parameter query-dependent GNN-based graph foundation model that incorporates node text semantics and augments an LLM with top-ranked nodes for answering.  Across six benchmarks (HotpotQA, MuSiQue, 2Wiki, and three G-bench variants), it reports consistent gains over strong graph-enhanced and embedding baselines, while also emphasizing scalability via mixed-precision training and distributed message passing.

Reviewers’ main concerns centered on (i) novelty being more “well-engineered unification” than conceptual leap, (ii) missing comparisons to advanced multi-step/agentic RAG and realism of the evaluation corpus, (iii) limited interpretability/case studies, and (iv) questions about reproduced baseline discrepancies (notably GFM-RAG).  The rebuttal added requested comparisons and larger-corpus experiments, introduced path-based interpretability, and clarified the GFM-RAG discrepancy with a documented bug fix and added results, which meaningfully strengthened the evidence for robustness and fair comparison.  Given the strengthened empirical support and the practical value of the unified abstraction + scalable training recipe, the AC's recommendation is accept.

**Reviewer Concerns:**

Addressed by rebuttal: jHd3’s requests for multi-step RAG comparisons and larger-corpus evaluation; qUas/gBkL’s interpretability request via gradient/path-based reasoning traces; and gBkL’s baseline reliability concern via added GFM-RAG-on-G-bench results and a clear explanation/fix for the evaluation mismatch.

Outstanding: novelty concerns are not fully eliminated (the main contribution is framing/unification/engineering rather than a new GNN primitive), and the work still does not comprehensively demonstrate Wikipedia-scale open-domain evaluation across all settings, so the “unified”/“general” claims should be framed with appropriate scope

**Reviewer Scores:**

I expect jHd3 would increase from 4 to 8, voRT would remain at 8, gBkL would revert from the temporary 4 to 6 after the clarified baseline reproduction and added comparisons, and qUas would sustain a score of 4 due to the  incremental novelty concern.

---

### Decision · Program_Chairs · 2026-01-26

Accept (Poster)